



**Quantification of the dust optical depth across spatiotemporal scales with the**
**MIDAS global dataset (2003-2017)**
Antonis Gkikas[1], Emmanouil Proestakis[1], Vassilis Amiridis[1], Stelios Kazadzis[2,3], Enza Di Tomaso[4],
Eleni Marinou[1,5], Nikos Hatzianastassiou[6], Jasper F. Kok[7] and Carlos Pérez García-Pando[4,8]
[1]Institute for Astronomy, Astrophysics, Space Applications and Remote Sensing, National Observatory of Athens,
Athens, 15236, Greece
[2]Physikalisch-Meteorologisches Observatorium Davos, World Radiation Center, Switzerland
[3]Institute of Environmental Research and Sustainable Development, National Observatory of Athens, Greece
[4]Barcelona Supercomputing Center, Barcelona, Spain
[5]Deutsches Zentrum für Luft- und Raumfahrt (DLR), Institut für Physik der Atmosphäre, Oberpfaffenhofen, Germany
[6]Laboratory of Meteorology, Department of Physics, University of Ioannina, Ioannina, Greece
[7]Department of Atmospheric and Oceanic Sciences, University of California, Los Angeles, CA 90095, USA
[8]ICREA, Catalan Institution for Research and Advanced Studies, Barcelona, Spain
Corresponding author: Antonis Gkikas (agkikas@noa.gr)
**Abstract**
Quantifying the dust optical depth (DOD) and its uncertainty across spatiotemporal scales is key to
understanding and constraining the dust cycle and its interactions with the Earth System. This study
quantifies the DOD along with its monthly and year-to-year variability between 2003 and 2017 at
global and regional levels based on the MIDAS (ModIs Dust AeroSol) dataset, which combines
MODIS-Aqua retrievals and MERRA-2 reanalysis products. We also describe the annual and
seasonal geographical distributions of DOD across the main dust source regions and transport
pathways. MIDAS provides columnar mid-visible (550 nm) DOD at fine spatial resolution (0.1° x
0.1°), expanding the current observational capabilities for monitoring the highly variable
spatiotemporal features of the dust burden. We obtain a global DOD of $0.032 \pm 0.003$ – approximately
a quarter ($23.4\% \pm 2.4\%$) of the global AOD – with about one order of magnitude more DOD in the
northern hemisphere ($0.056 \pm 0.004$; $31.8\% \pm 2.7\%$) than in the southern hemisphere ($0.008 \pm 0.001$;
$8.2\% \pm 1.1\%$) and about 3.5 times more DOD over land ($0.070 \pm 0.005$) than over ocean ($0.019 \pm$
$0.002$). The northern hemisphere monthly DOD is highly correlated with the corresponding monthly
AOD ($R^2=0.94$) and contributes 20% to 48% of it, both indicating a dominant dust contribution. In
contrast, the contribution of dust to the monthly AOD does not exceed 17% in the southern
hemisphere, although the uncertainty in this region is larger. Among the major dust sources of the
planet, the maximum DODs (~1.2) are recorded in the Bodélé Depression of the northern Lake Chad



Basin, whereas moderate-to-high intensities are encountered in the Western Sahara (boreal summer),
along the eastern parts of the Middle East (boreal summer) and in the Taklamakan Desert (spring).
Over oceans, major long-range dust transport is observed primarily along the Tropical Atlantic
(intensified during boreal summer) and secondarily in the North Pacific (intensified during boreal
spring). Our calculated global and regional averages and associated uncertainties are consistent with
some but not all recent observationally based studies. Our work provides a simple, yet flexible method
to estimate consistent uncertainties across spatiotemporal scales, which will enhance the use of the
MIDAS dataset in future studies.
**1. Introduction**
Mineral dust particles are emitted throughout the year across the arid and semi-arid regions of the
planet, when winds exceed a threshold velocity mainly determined by soil texture, soil moisture, and
surface roughness. While dust aerosols have mainly a natural origin, the contribution of
anthropogenic land use is estimated to be between 10% and 25 % (Tegen et al. 2004; Stanelle et al.,
2014; Ginoux et al., 2012). Dust is mobilized by microscale to synoptic scale phenomena, from dust
devils developed under strong surface heating (Koch and Renno, 2005), to "haboobs" formed by
intense cold-pool downdrafts related to deep moist convection (Knippertz et al., 2007), to synoptic
patterns associated with intensified pressure gradients (Klose et al., 2010) and low-level jets (LLJ;
Fiedler et al., 2013). Meteorology also plays a key role in the dust transport over maritime areas taking
place mainly across the Tropical Atlantic Ocean (Prospero and Mayol-Bracero, 2013; Yu et al., 2015),
the northern Pacific Ocean (Husar et al., 2001), the Mediterranean (Flaounas et al., 2015; Gkikas et
al., 2015), the Arabian Sea (Ramaswamy et al., 2017) and the southern Atlantic Ocean (Gasso and
Stein, 2007). Dust perturbs the radiation budget through direct (Sokolik and Toon, 1996), semi-direct
(Huang et al., 2006) and indirect (Haywood and Bucher, 2000) processes, leading to impacts upon
weather (Pérez et al., 2006; Gkikas et al., 2018; Gkikas et al., 2019) and climate (Lambert et al., 2013;
Nabat et al., 2015). Upon deposition, nutrient-rich dust particles can increase the productivity of
oceanic waters (Jickells et al., 2005) and terrestrial ecosystems (Okin et al., 2004) and perturb the
carbon cycle (Jickells et al., 2014). Dust has been associated with epidemics of meningococcal
meningitis in the African Sahel (Pérez García-Pando et al., 2014a, b) and with air quality degradation
in urban areas (Kanakidou et al., 2011) causing respiratory (Kanatani et al., 2010) and cardiovascular
(Du et al., 2016) disease when the population is exposed to high dust concentrations (Querol et al.,
2019). Other socio-economic sectors can be regionally affected by dust storms (Middleton, 2017),
including transportation (Weinzierl et al., 2012), agriculture (Stefanski and Sivakumar, 2009) and
solar energy production (Kosmopoulos et al., 2018).



Satellite measurements and numerical simulations have repeatedly shown the remarkable contrast
in dust load between the two hemispheres. The substantially higher dust load in the N. Hemisphere
is associated to the wider deserts extending across the so-called "dust belt" (Prospero et al., 2002;
Ginoux et al., 2012) in contrast to the smaller sources in Australia, South Africa and South America.
At global scale, most of the entrained dust loads in the atmosphere originate from tropical and sub-
tropical arid regions; yet, it is estimated that up to 5% of the global dust budget consists of particles
emitted from high-latitude sources (Bullard and Austin, 2011; Bullard et al., 2016). Given the key
role of dust aerosols in the Earth system it is imperative to monitor and understand the global dust
cycle along with its multi-scale spatiotemporal variability over long time periods and fine spatial
resolution. This task can be fulfilled to a certain degree using contemporary satellite instruments
providing accurate retrievals and global coverage over extended time periods. With this approach,
one of the key challenges is to discriminate dust from other aerosols. Several studies have combined
AOD and aerosol index (AI) (e.g., Middleton and Goudie, 2001; Prospero et al., 2002) or AOD, single
scattering albedo (SSA) and Angstrom exponent (AE) (Ginoux et al., 2012) to identify the most active
dust sources worldwide. Other studies have focused on the dust load and its variability in specific
regions such as the Atlantic Ocean and the Arabian Sea (Peyridieu et al., 2013), the Sistan basin
(Rashki et al., 2015), the Mediterranean (Gkikas et al., 2016), Europe and North Africa (Marinou et
al., 2017) and east Asia (Proestakis et al., 2018), among others. Liu et al. (2008) described the three-
dimensional structure of dust aerosols at global scale based on CALIOP vertically resolved retrievals
acquired during the first operational year of the CALIPSO satellite mission. A more advanced
approach has been introduced by Amiridis et al. (2013) and Marinou et al. (2017), who applied a
more realistic lidar ratio for the Saharan dust and a series of quality filters on the CALIOP vertical
profiles, in order to provide information about the vertical structure of dust layers at global scale and
coarse resolution in the LIVAS dataset (Amiridis et al., 2015). Ridley et al. (2016) quantified the
global average DOD and its uncertainty for the period 2004-2008 based on AOD retrievals from
passive spaceborne sensors (MODIS, MISR), ground-based (AERONET) and shipborne (MAN)
measurements from sun-photometers, and numerical simulations. Voss and Evan (2020) provided a
long-term DOD climatology over the Tropics and mid-latitudes at a coarse spatial resolution (1° x
1°) based on MODIS and AVHRR observations, where DOD was estimated based on: AOD, SSA
and AE over land following Ginoux et al. (2012) and AOD, fine and coarse AOD (AERONET) and
MERRA-2 winds over ocean. Based on vertically-resolved CALIOP retrievals and columnar MODIS
optical properties, Song et al. (2021) provided a long-term 4D global dust optical depth dataset,
excluding the polar regions, over the period 2007 – 2019. In their approach, they took advantage of
spaceborne observations that can be used for the discrimination/identification of dust aerosols
characterized by their aspherical shape, coarse size and absorption.





Our study provides a global and regional quantification and description of the DOD based on the
new ModIs Dust AeroSol (MIDAS) dataset (Gkikas et al., 2021). The powerful and innovative
elements of the MIDAS DOD dataset are the: (i) daily availability and fine spatial resolution (0.1° x
0.1°), (ii) full global coverage including the sources and downwind areas (both over land and sea),
(iii) 15-year temporal range (2003 – 2017) using the most updated MODIS data collection, (iv) grid-
cell level uncertainty quantification. In this contribution, we first describe the annual and seasonal
geographical distribution of DOD across the main dust source regions and transport pathways
(Section 4.1). We then quantify the average DOD and its monthly and year-to-year variability at
global, hemispherical and regional levels, along with its fractional contribution to the AOD (Section
4.2). We summarize the main findings in Section 5.
**2.  ModIs Dust AeroSol (MIDAS) dataset**
Our study is based on the MIDAS global fine resolution dataset described in detail in Gkikas et
al. (2021). We analyse the DOD at 550 nm at 0.1° x 0.1° spatial resolution between 2003 to 2017.
The MIDAS DOD results from the combination of the quality-filtered MODIS aerosol optical depth
(AOD, Collection 6.1, Level 2; Levy et al., 2013) and the MERRA-2 (Modern-Era Retrospective
Analysis for Research and Applications, version 2; Gelaro et al., 2017) fraction of AOD that is due
to dust (MDF). In Gkikas et al. (2021), the MDF was evaluated against the dust fraction obtained
from quality-assured dust and non-dust CALIOP (Cloud-Aerosol Lidar with Orthogonal Polarization;
Winker et al., 2009) profiles, available from the LIVAS database (Amiridis et al., 2015; Marinou et
al., 2017; Proestakis et al., 2018). The MDF compares well with the LIVAS dust fraction over the
dust-abundant areas extending across the NH dust belt, with maximum underestimations of 10 % in
Asian deserts. The agreement is more limited in North America and the Southern Hemisphere
(Figures 1 and 2 in Gkikas et al., 2021). Overall, the MIDAS DOD is well correlated with AERONET
dust-dominant retrievals (R=0.89 at global scale) and the absolute biases are mainly below 0.12 at
stations near sources (Figures 3 and 4 in Gkikas et al., 2021). The MIDAS DOD dataset was further
verified against the LIVAS DOD and compared with MERRA-2 DODs (Figure 5 in Gkikas et al.,
2021). Among the three datasets, there is good agreement on the monthly variability of the global and
hemispherical DODs as well as on their long-term averages (Figure 6 and Table 1 in Gkikas et al.,
2021). Moreover, the annual and seasonal DOD patterns are broadly similar in the three datasets
throughout the period 2007 – 2015. Nevertheless, regionally differences are found due to the different
techniques (passive and active remote sensing, numerical simulations) applied for the DOD
derivation.






**3. Spatiotemporal averaging and propagation of grid-cell level uncertainties**
In section 4.2 we provide DOD estimates that are averaged in space (regionally and globally) and
in time (over months, seasons and years) along with their respective uncertainties. Averaging is
performed according to the upper branch of Figure 5 in Levy et al. (2009), i.e. spatial averaging is
performed after grid cell temporal averaging for any of the timescales considered. The uncertainties
of the DOD averages at the different spatiotemporal scales are based on the propagation of the daily
grid cell uncertainties provided within the MIDAS dataset and presented in Gkikas et al. (2021). In
short, the daily grid cell uncertainties combine the uncertainties of the MODIS AOD and the
MERRA-2 MDF with respect to AERONET and LIVAS, respectively. The former is based on linear
equations expressing the uncertainty with respect to AERONET AOD over the ocean (Levy et al.,
2013) and land (Levy et al., 2010; Sayer et al. 2013) with updated coefficients for C061 data
depending on vegetated and arid surface types (see equations 4 to 7 in Gkikas et al., 2021). The latter
is based on a quartic (fourth degree) polynomial equation expressing the uncertainty with respect to
the LIVAS dust fraction (see equation 8 in Gkikas et al., 2021).
In order to estimate the uncertainties of the spatiotemporal averages we first assume that each of
the daily grid cell uncertainties are composed of (1) a fraction that is completely random in time and
space, (2) a fraction that is systematic (correlated) in time and random in space and (3) a fraction that
is systematic (correlated) in space and random in time. Our framework also assumes that the fraction
of the daily grid cell uncertainty that is correlated both in space and time, for instance an instrument
bias, is very small and therefore neglected. Under this framework, the propagation of uncertainty
fraction (1) is negligible across the spatiotemporal scales considered, the propagation of uncertainty
fraction (2) depends upon the size of the domain considered but is negligible at global scale and across
most of the regional domains considered in this study, and propagation of fraction (3) accounts for
most of the total average uncertainty. Since we cannot know fractions (1), (2) and (3) and (1) and (2)
are negligible or small, we assume that (3) represents 100 % of the uncertainty, i.e the grid cell
uncertainty is systematic (correlated) in space and random in time, to provide an upper limit on the
uncertainty. In addition, we also take into account the sampling uncertainty when temporally
averaging over each grid cell using the standard error, i.e., we take the standard deviation divided by
the square root of the number of measurements.
In practice, when averaging the daily values for every grid cell $i$ over months, seasons, or years,
the uncertainty $\sigma'_i$ is obtained by adding in quadrature the daily uncertainties $\sigma^2_{N_i}$ and dividing by the
number of available daily measurements $N_i$:
$$\sigma'_i = \frac{\sqrt{\sigma^2_{i,1} + \sigma^2_{i,2} + \cdots + \sigma^2_{N_i}}}{N_i} \textbf{(Eq. 1)}$$




In addition, we add in quadrature $\sigma'_i$ and the standard error $SE_i$ to obtain the total uncertainty of
the temporal average $\sigma_i$ for every grid cell:

$$\sigma_i = \sqrt{\sigma'^2_i + SE^2_i} \text{ (Eq. 2)}$$

$$SE_i = \frac{SD_i}{\sqrt{N_i}} \text{ (Eq. 3)}$$


where $SD_i$ is the standard deviation of the daily values in grid cell $i$. The standard error measures how
far the sample mean could be from the true population mean.

Finally, when spatially averaging globally or regionally, under the assumption that the errors are

correlated across space, the overall uncertainty is calculated by averaging $\sigma_i$ across the $N_j$ grid cells
in spatial domain $j$ weighted by the grid cell area fraction with respect to the total area (i.e., grid cell
/ total area = w$_i$) with available retrievals:

$$\sigma_j = \sum_{i=1}^{N_j} w_i * \sigma_i \text{ (Eq.4)}$$


**4. Results**

Our analysis is divided in two main parts. In the first one (Section 4.1) we assess the annual and

seasonal climatological DOD maps for nine distinct regions. In the second one (Section 4.2),
emphasis is given on the quantification DOD averages along with their monthly and interannual
variability fractional contribution to the AOD, from a global to hemispherical level as well as for
specific regional domains.

*4.1 Annual and seasonal geographical distributions of DOD*

*4.1.1 North Africa, Tropical Atlantic Ocean and Mediterranean*

According to the long-term average map (Fig. 1), the maximum DODs (up to 1.2) are recorded in

the Bodélé depression, which is considered the most active individual dust source of the planet
(Washington et al., 2003; Koren et al., 2006; Ginoux et al., 2012). Over the area, the prevailing strong
winds are intensified further between the Tibesti mountains and the Ennedi ridge (Washington et al.,
2009) forming a low-level jet (Washington and Todd, 2005). This dominant wind pattern, affected
by the local topography (Washington et al., 2009), acts as the driving force mobilizing mineral
particles from arid and erodible soils of the region (Tegen et al., 2006). Under these favorable
conditions, dust aerosols are easily uplifted and accumulated in the atmosphere thus causing the very





high DODs (> 0.5) observed in the broader area (Chad, Niger). Throughout the year, the high DOD
levels are quite persistent exhibiting, however, a seasonal variation with more intense loads recorded
during DJF (Fig. S1-i) and MAM (Fig. S1-ii) following the annual cycle of source activation
(Washington et al., 2009). The second hotspot in N. Africa is situated between the northern parts of
Nigeria and the southern parts of Niger with annual DODs reaching up to 0.7 (Fig. 1) while on
seasonal basis vary from 0.4 (SON; Fig. S1-iv) to 0.8 (JJA; Fig. S1-iii). MIDAS DODs match well
with those presented by Rajot et al. (2008), who relied on ground-based sunphotometric
measurements of AOD obtained at the Banizoumbou AERONET site. Very high DODs are also
evident along the coasts of the Gulf of Guinea, which may be unrealistic considering that dust aerosols
are mainly transported there and are mixed with anthropogenic and biomass burning (Knippertz et
al., 2015). Along this area of high DODs, MERRA-2 also overestimates the dust fraction compared
to LIVAS (Gkikas et al., 2021) thus resulting in higher intensities according to the applied
methodology (Section 2). Moreover, the temporal availability of DODs in the region is very limited
(<10%; Fig. 8-c in Gkikas et al., 2021), the DOD uncertainty is large and AOD outliers, either realistic
or cloud contaminated, can yield exceptional high DODs in this complex environment where aerosol
and clouds are spatially correlated (Andrew Sayer, personal communication).

Across the Sahara Desert, there is a distinct longitudinal contrast with more intense dust loads in

western North Africa than in eastern North Africa (Fig. 1). In the former sector, the DODs range
mainly from 0.3 to 0.6 while over the eastern parts of the Sahara the corresponding limits are bounded
between 0.1 and 0.3 without revealing significant intra-annual variation. During MAM (Fig. S1-ii),
along the southern Sahel, the activation of dust sources results in DODs which locally can exceed
0.8, while during boreal summer (Fig. S1-iii) a vast area of the western Sahara is under the impact of
heavy dust loadings (DOD > 0.5). According to Ginoux et al. (2012), in the former region, dust is
mainly produced by agricultural activities (cultivation, overgrazing) disturbing soils in which alluvial
sediments have been accumulated. Northwards, dust has natural origin and the accumulation of
mineral particles is favored by the development of the Saharan Heat Low (SHL) affecting also the
prevailing airflow (harmattan winds) as well as the West African Monsoon (WAM) (Schepanski et
al., 2017). Under these meteorological conditions, several dynamic processes, from mesoscale to
microscale, are taking place triggering dust emission (Knippertz and Todd, 2012) from highly active
sources (Schepanski et al., 2007).

Under the impact of the trade winds, Saharan dust can travel across the tropical Atlantic Ocean

reaching the Caribbean Sea, the southern United States and northeastern South America (Prospero,
1999; Prospero et al., 2014). The signal of this long-range transport is evident on the annual
climatological pattern (Fig. 1) with DODs up to 0.6 (off the western Saharan coasts) fading down to
0.1 at the maximum distance. Within the course of the year, the Saharan dust plume varies in terms



of intensity, range and latitudinal position, as it is depicted in Figure S1. During boreal summer (Fig.
S1-iii), the corridor of the transatlantic dust transport is bounded between 10° N and 20° N latitudes
whereas both the intensity (DODs up to 0.6) and the range are maximized. During boreal winter (Fig.
S1-i), the dust zone migrates southwards (between Equator and 10° N) while maximum (up to 0.6)
and considerable (0.1-0.2) DODs are observed over the Gulf of Guinea and mid-Atlantic (45° W),
respectively. Between the transition seasons (Fig. S1-ii, S1-iv), dust loads are stronger in MAM
(~0.45), mainly residing within 5° N and 20° N latitudes, in contrast to SON (~0.3) when are shifted
northwards (10° N and 25° N). According to the existing literature, several factors modulate the
westwards propagation of dust plumes, originating in the western Sahara and the Bodélé Depression,
over the tropical Atlantic. For instance, the south-north displacement of the Saharan plumes is driven
by the location of the Intertropical Convergence Zone (ITCZ) and the disturbances of the African
easterly jet (Knippertz and Todd, 2012; Doherty et al., 2012). Teleconnection patterns, such as the El
Niño–Southern Oscillation (ENSO; Prospero and Lamb, 2003), the North Atlantic Oscillation (NAO;
Ginoux et al., 2004) and the North African Dipole Index (NAFDI; Rodríguez et al., 2015) have been
also studied in order to interpret the decadal variations of dust concentrations over the Atlantic.
Likewise, the vegetation coverage across the Sahel as well as the wind speeds, determined by the
prevailing atmospheric circulation, over the Sahara play a key role on the amount of the emitted dust
particles.

Due to the vicinity of the largest deserts of the planet, the Mediterranean is affected by dust

outbreaks throughout the year (Gkikas et al., 2013; Marinou et al., 2017). Mineral particles originating
primarily from north African and secondarily from Middle Eastern deserts are transported towards
the Mediterranean under the prevalence of cyclonic systems (Gkikas et al., 2015). The intensity of
dust loads decreases for increasing latitudes, forming a distinct south-north gradient with DODs up
to 0.20 between the gulfs of Gabes (Tunisia) and Sidra (Libya), according to the annual pattern (Fig.
1). Among seasons (Fig. S1), DODs vary on the locations where the maximum levels are recorded as
well as on their magnitude, attributed to the position of the prevailing synoptic systems (Gkikas et
al., 2015). The central and eastern Mediterranean sectors are affected by dust loads mainly in spring
(DODs up to 0.3; Fig. S1-ii) and winter (DODs up to 0.12; Fig. S1-i). In summer (Fig. S1-iii), dust
activity is more pronounced in the western parts with optical depths up to 0.18 (Alboran Sea), while
thanks to the fine resolution product, "hotspots" of similar DODs can be identified in the southern
parts (Andalucia) of Spain. In SON (Fig. S1-iv), dust loads are found in the central Mediterranean
with DODs lower than 0.12 off the Tunisian and Libyan coasts.







*4.1.2   Middle East*

In the Middle East, there is a zone of moderate-to-high DODs (locally up to 0.8) extending from
Mesopotamia to the southern parts of the Saudi Arabia, where one of the largest sand deserts of the
world (Rub' al Khali) (Hamidi et al., 2013) is situated (Fig. 2). Based on Ginoux et al. (2012), the
origin of mineral particles between Tigris and Euphrates as well as across the Rub' al Khali Desert is
mainly natural while in the intermediate part (Ad-Dahna Desert) dust accumulation is attributed to
the mixing of anthropogenic and hydrological sources. Slightly higher maximum DODs (up to 0.7;
Fig. 2) are recorded in Oman and particularly between Dhofar and Al Wusta, in contrast to previous
studies (Pease et al., 1998) which have identified the Wahiba Sands area as a major dust source or the
coastal areas of Yemen (Ginoux et al., 2012). On a seasonal basis, the intensity of mineral loads
exhibits a strong variability with minimum DODs (up to 0.4) during DJF (Fig. S2-i) and SON (Fig.
S2-iv) and maximum (up to 1) during the dry period of the year (Figs S2-ii, S2-iii), being in agreement
with the results presented in Yu et al. (2013). More specifically, across the Arabian Peninsula, the
increase in DOD levels is getting evident in boreal spring and it is further intensified during summer
months. Dust storms emanating in Iraq and the eastern parts of Saudi Arabia favor dust transport
towards the Persian Gulf (Gianakopoulou and Toumi, 2012) account for the considerable high DOD
levels (>0.6) found there. Due to convergence of the northern-northernwesterly Shamal winds (Yu et
al., 2016) and the airflow from the subtropical anticyclone, in JJA, mineral particles are travelling at
even longer distances towards the northern Arabian Sea (Ramaswamy et al., 2017), as indicated by
the intense dust loads (DODs up to 0.5; Fig. S2-iii) contributing about half of the AOD (Jin et al.,
2018). Likewise, during boreal summer, short-range dust transport takes place off the coasts of Oman
and Yemen (Gulf of Aden). Among seas in the vicinity of the Arabian Peninsula, the most intense
dust loads are observed in the Red Sea, forming a clear latitudinal gradient on annual (Fig. 2) and
summer (Fig. S2-iii) geographical DOD patterns, as it has been noted also in Brindley et al. (2015)
and Banks et al. (2017). Due to its location, the southern sector of the Red Sea receives dust aerosols
either originating from the Republic of Sudan or from the Arabian Peninsula, depending on the zonal
airflow (Banks et al., 2017). Dusty air masses travelling westwards are uplifted when they are
crossing the mountain range in the southwestern Arabian Peninsula and for this reason dust loads
over the southern basin are suspended above 2 km (Banks et al., 2017). On the contrary, low-elevated
dust layers are recorded when winds blow from west, triggering dust emission from the Tokar Gap
(Sudanese coasts) and subsequently dust outflows into the southern Red Sea (Banks et al., 2017).





### 4.1.3 Central and southwest Asia

Northwards and eastwards of the Caspian Sea, various deserts are situated in the central segments of the Asian continent. Most part of Turkmenistan is occupied by the Karakum Desert while the Kyzylkum Desert is located in Uzbekistan. Other arid regions stretch between the Caspian and Aral Seas (Ustyurt plateau), in the eastern and southern flanks of the Aral Sea (Solonok Desert) and in the lowlands of western Kazakhstan and southeastern Russia (Ryn Desert) (Elguindi et al., 2016). Based on our seasonal spatial patterns (Fig. S3), the major dust activity is recorded in the Ustyurt Plateau (Li and Sokolik, 2018) and in the large lagoon embayment of Garabogazkol (Shen et al., 2016), a gulf of Turkmenistan dried into a salt-covered playa (Gills, 1996), with minimum (in DJF and SON) and maximum (in MAM and JJA) DODs equal to ~0.2 and ~0.4, respectively. In the rest of areas, the corresponding upper limits can reach up to 0.8-0.9, during boreal summer, in localized spots (Chimboy Lake, Sarygamysh Lake) across the Karakum and Kyzylkum Deserts. For the same season, moderate dust loadings (DOD up to 0.25) are encountered in the southern Caspian Sea (Elguindi et al., 2016) as the result of transported mineral particles mainly coming from the sandy deserts of Turkmenistan (Xi and Sokolik, 2015), under the impact of eastern/southeastern winds (Shen et al., 2016). Since the 1960s, the anthropogenic intervention (agricultural activities, over-irrigation) caused the retreat of the Aral Sea and the formation of the Aralkum Desert (Saiko and Zonn, 2000; Micklin, 2007) from which large amounts of aeolian dust are emitted and travel distances of hundreds of kilometers (Indoitu et al., 2015). According to the annual climatological map (Fig. 3), extremely high DODs (> 1) are found in the southeastern parts of the Aralkum Desert (Fig. 3) which are also persistent among the seasons (Fig. S3).

In the Sistan basin, extending between Iran-Pakistan-Afghanistan, the long-term average JJA DODs can reach up to 1.1 (Figure S3-iii) in the Margo Desert (Afghanistan), due to the frequent occurrence of dust storms (Middleton, 1996), triggered by the northerly Levar winds, blowing from June to September (Alizadeh Choobari et al., 2014). These maximum DOD levels are substantially higher than the annual mean (0.8; Figure 3) as well as against the corresponding averages for the other seasons. Thanks to the high-resolution MIDAS DOD, we identify the borders of other active arid regions, surrounded by mountain ranges, such as the Rigestan (Afghanistan), the Balochistan (Pakistan), the Dasht-e-Kavir (Iran), the Dasht-e-Lut (Iran) and the Jazmurian drainage basin (Iran). In the aforementioned topographic lows, the magnitude of the dust loads is significantly lower than those observed in the Margo Desert and can be as large as 0.6 (Balochistan) during hot-dry months (Figure S3-iii). The presence of absorbing mineral particles, over the area and in the northernmost part of the Arabian Sea, is also confirmed by the high AI values, especially in June-July, discussed





by Rashki et al. (2015), who relied on long-term records obtained by the OMI and TOMS spaceborne
sensors.

*4.1.4    Indian subcontinent*

In the Indian subcontinent, the maximum annual DODs (~0.5; Fig. 4) are observed along the Indus

river basin, in the western side of the Thar Desert whereas a branch of gradually decreasing DODs,
along the Indo-Gangetic plain towards eastwards directions, is also evident. Ginoux et al. (2012)
stated that much of dust activity in the Indus river basin is attributed to the suspension of soil particles
originating primarily from agricultural land use and to a lesser extent from the desiccation of
ephemeral water bodies. The strong presence of absorbing coarse particles over the area is further
supported by the coexistence of considerably high Aerosol Index (AI) values (Alam et al., 2011). As
indicated by the seasonal patterns (Fig. S4), the processes regulating the suspended dust loads are
highly variable during the year causing a remarkable temporal variability of DOD, which is low (<0.3)
in DJF and SON, moderate in MAM (<0.5) and maximum in JJA (<0.8). Similar seasonal variability
is evident in the Thar Desert, in agreement with the findings of Proestakis et al. (2018) and Dey and
Di Girolamo (2010), who used vertically-resolved (CALIOP) and multi-angle (MISR) satellite
retrievals, respectively. Nevertheless, our climatological DODs are higher with respect to the
CALIOP corresponding values and the MISR non-spherical AODs, particularly when dust activity
over the area is pronounced. During the pre-monsoon season, westerly to northwesterly winds are
blowing over the Thar Desert mobilizing dust particles which subsequently are advected towards the
Indo-Gangetic basin (Dey et al., 2004; Srivastava et al., 2011). According to our results, between the
Haryana state and the eastern parts of the plain, DODs fade down from ~0.6-0.7 to ~0.1-0.2, forming
a NW-SE gradient (Figs. S4-ii, S4-iii). Such high DODs are attributed to the eastwards propagation
of intense dust storms having a strong signature on the optical, microphysical and radiative properties
derived by AERONET stations operating in the region (Prasad et al., 2007a; Prasad et al., 2007b; Eck
et al., 2010).

*4.1.5    East Asia and North Pacific Ocean*

Northwards of the Tibetan Plateau is located the Tarim Basin (northwest China) in which one of

the largest natural dust source resides, the Taklamakan Desert. This elevated desert area (average
elevation 1.1 km) is surrounded by the Pamir Plateau (average elevation 5.5 km) in its west side, by
the Kunlun Shan range (average elevation 5.5 km) in its southern flanks and by the Tian Shan range
(average elevation 4.8 km) along its northern boundaries while only in its eastern margin the ground





elevation is low (Ge et al., 2014). DODs are maximized in spring (Fig. S5-ii) yielding values up to 1
along the foothills of the Tian Shan and Kunlun Shan ranges, attributed to the role of the topography
on winds strengthening (Ge et al., 2014). Similar values are recorded in JJA (Fig. S5-iii) but the
geographical distribution reveals that the highest DODs are less widespread in contrast to spring.
Throughout the year, the weaker dust loads are recorded during winter and autumn. Our results are
consistent with relevant studies that rely on active and passive satellite retrievals either of pure dust
load (Proestakis et al., 2018) or AOD (de Leeuw et al., 2018; Sogacheva et al., 2018).

A common feature in the seasonal DOD patterns is the reduction of dust loads' intensity towards

the interior parts of the Taklamakan Desert, as it has been also documented by Ge et al. (2014), who
utilized MISR retrievals. The high-resolution of the MIDAS DOD dataset provides in detail the
spatial information of these geographical patterns. During spring, similar high DODs to those found
over the Taklamakan Desert are recorded in the Qaidam Basin (northeast side of the Tibetan Plateau),
surrounded by the Atlun, Kunlun, Qilian mountain ranges, attributed to strong downslope winds
causing the erosion of soil particles (Rohrmann et al., 2013) and their entrainment into the
atmosphere. The intensity of dust loads over the Gobi Desert (north China – south Mongolia) hardly
exceeds 0.3 on an annual basis (Fig. 5) while it can reach up to 0.4 during spring (Fig. S5-iii). The
remarkable deviations in dust abundance between Taklamakan and Gobi during springtime are
interpreted by variations in soil characteristics. More specifically, Taklamakan is composed mainly
by fine sand particles in contrast to the rocky soils of the Gobi Desert (Sun et al., 2013). Due to these
differences in soil textures, dust particles from the former desert region can be emitted even with low
wind speeds while they are uplifted at higher elevations in the troposphere, as it has been shown with
MISR stereo observations (Yu et al., 2019) and CALIOP lidar profiles (Proestakis et al., 2018). The
injection of Taklamakan dust particles at higher altitudes increase their residence time inducing also
their entrainment into the upper-level westerly airflow, around at 4 a.m.s.l., both contributing to the
higher potential for long-range transport (Yu et al., 2019), in contrast to Gobi dust, towards the
continental E. Asia and the northern Pacific Ocean. Under the impact of cold fronts, propagating
eastwards (Eguchi et al., 2009) in spring, air masses carrying mineral particles, during the first two
days of dust transport, affect a wide area of China (Yu et al., 2019), from near sources to its eastern
parts, through the Hexi Corridor and the Loess Plateau (DODs ranging from 0.2 to 0.4; Fig. S5-iii).
Subsequently, the Asian dust plumes are suspended over the Yellow Sea, the Korean Peninsula and
further eastwards, in a latitudinal band bounded between the parallels 30°N and 45°, reaching the
west coasts of the United States (Yu et al., 2008). Across this "belt", where the Trans-pacific dust
transport is taking place, the springtime DODs decrease smoothly from 0.15 to 0.05 (Fig. S5-ii). In
summer (Fig. S5-iii), DODs up to 0.05 are observed between 40° N and 60° N indicating a northwards



displacement of the Asian dust layers (mainly originating from the Gobi Desert) due to the weakening
and northwards shift of the polar jet streams (Yu et al., 2019).

*4.1.6   North America*

Across N. America, the major dust activity is detected in southwest United States and in northwest

Mexico with annual and seasonal DODs hardly exceeding 0.15, as illustrated in Figures 6 and S6,
respectively. These weak dust load intensities are mainly recorded in the Sonoran and the Mojave
Deserts while lower values are found in the Chihuahuan Desert in which isolated spots (e.g. White
Sands Desert) become visible thanks to the high-resolution of the MIDAS DOD dataset. Low-to-
moderate DODs are evident in the Great Plains with local maxima (exceeding 0.2 in spring; Fig. S6-
ii) in the Great Salt Lake Desert and in the surrounding area as well as in the Baja Californian Desert
(Mexico; DODs up to 0.14), residing in the western side of the Gulf of California. Our annual spatial
distribution of DOD (Fig. 6) is highly consistent with those of frequency of observation (FoO) of
DOD (Ginoux et al., 2012; Baddock et al., 2016) and AI given by Prospero et al. (2002). Moreover,
the increase of dust loads' concentration in MAM (Fig. S6-ii), has been also documented by Hand et
al. (2016) and Tong et al. (2017), both relying on aerosol observations acquired at numerous stations
of the Interagency Monitoring of Protected Visual Environments (IMPROVE) network. During
springtime, dust emission over the broader area is associated with the transmit of Pacific cold fronts
inducing dust-entraining winds as the result of pressure gradient enhancement (Rivera Rivera et al.,
2009). The geomorphological soil characteristics are determinant for dust emission with the most
prominent natural sources being ephemeral and dry lakes (Baddock et al., 2016) while anthropogenic
dust aerosols are mainly emitted in the Great Plains and in the eastern side of the Gulf of California
(Ginoux et al., 2012).

*4.1.7   Australia*

Earlier studies based on unconstrained numerical simulations (Tanaka and Chiba, 2006; Wagener

et al., 2008) have shown that among the desert areas of the S. Hemisphere, the largest contribution of
dust particles arises from Australia. However, a more recent assessment (Kok et al., 2021b) in which
dust models have been constrained by observations revealed that the emitted dust amounts from S.
America are slightly higher than those of Australia. Due to the fairly bright landmasses and the
predominance of weak aerosol loadings, there is minimal contrast between surface and atmosphere
leading to systematic algorithm uncertainties, which can explain the slightly lower land DODs than
those recorded in the surrounding oceanic regions (Fig. 7 and S7). Nevertheless, in the sources as



well as in areas affected by dust plumes the atmospheric signal becomes evident. In particular, the
highest dust emissions are encountered in the Lake Eyre Basin (LEB; Prospero et al., 2002) composed
by ephemeral lakes, alluvial channels, gibber (stone-covered plains), aeolian sand deposits and
bedrocks (Bullard et al., 2008). Based on the annual climatological pattern (Fig. 7), DODs can locally
exceed 0.2 (in the southern parts) but in general vary between 0.06 and 0.12. From a seasonal
perspective (Fig. S7), the highest DODs (mainly up to 0.18 in the Warburton River estuary, few
exceedances above 0.4 are found in local spots) are recorded during austral summer (DJF; Fig. S7-i)
and spring (SON; Fig. S7-iv). Similar seasonal variation in ground-based sunphotometric
observations at nearby sites (Birdsville, Tinga Tingana), with slightly lower AODs, has been reported
by Mitchell et al. (2017). Southwards of the LEB, three spots of notable DODs (up to 0.2 in SON;
Fig. S7-iv) are identified in the Lakes Gairdner, Torrens and Frome while northeastwards (Lake
Yamma Yamma) and northwards (Simpson Desert) from the basin the suspended dust loads exhibit
optical depths as large as 0.12 during the driest months of the year. Similar maximum DODs are
recorded in the Northern Territory and in the western side of the Great Dividing Range (Queensland)
and in contrast to Ginoux et al. (2012) these levels appear in DJF instead of SON. In the southwestern
coastal parts of the Australian landmass as well as in Riverina (southeast), during austral spring (Fig.
S7-iv) very low DODs are evident associated with anthropogenic dust originating from agricultural
activities (Ginoux et al., 2012). Finally, during the same season, weak signals (DODs up to 0.05) of
dust transport are revealed over the Tasman and Timor Seas attributed to the eastward movement of
cyclonic frontal systems causing the entrainment of mineral particles in air masses that can travel at
long distances (Knight et al., 1995; Choobari et al., 2012).

*4.1.8   South Africa*

Dust activity in S. Africa is mainly related with short-range and short-lived plumes (Vickery et

al., 2013) that are suspended at low tropospheric altitudes (below 600 hPa) due to the predominance
of anticyclonic circulations inhibiting the vertical extension of dust layers (Piketh et al., 1999).
Mineral aerosol loadings are mainly originating from the ephemeral lake basins of the Etosha Pans
(Namibia) and Makgadikgadi Pans (Botswana) and the Namib Desert (Bryant et al., 2007; Vickery
et al., 2013). In the aforementioned source areas, the maximum annual (Figure 8) and seasonal (Figure
S8) DODs are equal to 0.1 and 0.16, respectively. Throughout the year, the increase of DODs in
Etosha and Makgadikgadi Pans is evident primarily in DJF (Figure S8-i) and secondarily in SON
(Figure S8-iv). Our results are consistent with those provided by Ginoux et al. (2012) and Bryant et
al. (2007) for the former region (including also the Kalahari Desert in which very weak dust loads are
recorded), contradictory for the latter one and opposite with the findings of Vickery et al. (2013) for





both sources. In these arid areas dust emission is linked with lakes' inundation, characterized by
strong intra-annual variability, playing an important role when different time periods are considered.
However, it must be also taken into account the moderate performance of the MERRA-2 dust portion
with respect to LIVAS in S. Africa as well as in most desert areas of the S. Hemisphere (Gkikas et
al., 2021). Along the Namibian coastline, the deviations of DOD between the high- and low-dust
seasons are small indicating that dust activity remains relatively constant within the course of the year
(Ginoux et al., 2012). Soil particles from salt pans and dry river beds of the Namib Desert are emitted
from aeolian processes related to bergwinds (katabatic winds) blowing in the escarpment, from the
Central Plateau down to the coasts (Eckardt and Kuring, 2005). Dust outflow towards the Southern
Atlantic Ocean, with a SE-NW orientation, it is shown between 18° S and 9° S during austral winter
(DODs up to 0.08; Fig. S8-iii), becoming more evident in SON (Fig. S8-iv), being in agreement with
the geographical distributions provided by Voss and Evan (2020). Such transport is favored by the
propagation of barotropic low-level easterly waves formed between continental high pressure systems
and the semi-permanent South Atlantic anticyclone (Tyson et al., 1996). Finally, weak signals of
DODs are recorded in the croplands north of Cape Town, with annual and DJF DODs not exceeding

0.1.


*4.1.9   South America*

In South America, the most intense dust loads are encountered in the Patagonia Desert where the
most active dust sources are situated in the river basins of the Rio Negro and Chubut provinces and
in its southern end. Among these areas, higher DODs (up to 0.16 in DJF; Figure S9-i) are found along
the Rio Negro attributed to anthropogenic dust originating from overgrazing, irrigation and oil
prospecting (McConnell, et al., 2007; Mazzonia and Vazquez, 2009). In southern latitudes, mineral
particles originate from glacier washout plains (Hernández et al., 2008). Under favorable
meteorological conditions, aeolian dust from Patagonia travels either towards the southern Atlantic
Ocean, contributing to iron concentrations and marine biological productivity in the surface waters
(Johnson et al., 2011), or towards the Antarctica peninsula (Gassó et al., 2010), as it has been found
in ice core samples (Basile et al., 1997). Both transport pathways are not visible in our climatological
patterns (Figures 9 and S9) since dust outbreaks are not so strong (Foth et al., 2019) while the
extended cloud coverage over the region results in large observational gaps of the spaceborne
retrievals (Gassó and Torres, 2019). Along the western side of Andes, dust emission arises from
natural sources located in the Sechura (Peru), Nazca (Peru) and Atacama (Chile) Deserts (Ginoux et
al., 2012). In the aforementioned regions, the annual DODs (Figure 9) can reach up to 0.1, 0.08 and
0.06, respectively, while the intra-annual variability is characterized weak (Figure S9). During MAM




(Figure S9-ii), DODs up to 0.16 appear in Guyana, Suriname and French Guiana as well as over their
offshore areas while similar intensities are evident in the northern parts of the Amazon rainforest
(around the Equator and bounded between 65°W and 60°W). The presence of coarse mineral particles
(Moran-Zuloaga, et al., 2018) over these distant areas from deserts, is attributed to the long-range
dust transport from North Africa across the Atlantic Ocean (Yu et al., 2015), under the impact of the
trade winds, taking place northwards of the convective precipitation zone formed around the ITCZ.
Finally, the latitudinal zone of weak DODs in the western parts of Brazil, fading down abruptly
eastwards of ~58° W, indicates an artifact of the MIDAS product that becomes more evident in SON
(Fig. S9-iv). This peculiar pattern is induced by the MERRA-2 dust fraction (results not shown here)
which is used for the derivation of MIDAS DOD from the MODIS AOD. An additional deficiency is
the relatively large DODs over an area where biomass burning particles, emitted at enormous amounts
by extended wildfires, clearly dominate over other aerosol species. Under these conditions, the non-
dust AODs are very high as well as their relevant uncertainties (Eqs. 5-7 in Gkikas et al. (2021)) while
the reliability of the MERRA-2 dust fraction downgrades there (see Fig. 2 in Gkikas et al. (2021)).

*4.2  DOD averages and variability at global, hemispherical and regional scales*

In this section, we discuss the average AOD and DOD along with their monthly and interannual
variability at global, hemispherical and regional scales. The left column of Figure 10 shows the
interannual timeseries of AOD (black curve) and DOD (red curve) averaged over the whole globe
(upper panel; GLB), the Northern Hemisphere (middle panel; NHE) and the Southern Hemisphere
(bottom panel; SHE). The right column of Figure 10 depicts the monthly seasonal cycle of AOD and
DOD along with the DOD-to-AOD ratio (blue curve) while the shaded areas correspond to the total
uncertainty.
The significant role of dust particles in the global aerosol budget becomes evident by visually
inspecting the AOD and DOD interannual timeseries (Fig. 10 i-a). The monthly contribution of
suspended dust to the total AOD varies from 14% to 39%, with minimum values mainly in DJF and
maximum values in MAM or JJA depending on the year. Monthly DODs range from $0.016 \pm 0.013$
(Dec 2005) to $0.063 \pm 0.028$ (Mar 2012), whereas the long-term global annual average is equal to
$0.032 \pm 0.003$ (Table 1). The global DOD mean, computed here from the fine resolution data, is
almost identical with those obtained by the coarse spatial resolution MERRA-2 and MIDAS DODs
and slightly higher than those calculated based on LIVAS-CALIOP (0.029) (see Table 1 in Gkikas et
al. (2021); it is noted the three datasets they had been collocated). Likewise, our global average and
uncertainty computed over the period 2004-2008 ($0.033 \pm 0.004$) is close to the one obtained in Ridley
et al. (2016) ($0.030 \pm 0.005$), despite the different methods applied for the derivation of DOD and its





uncertainty. Our global DOD long-term average is very close to the CALIOP derived value (0.029)
and about half of the MODIS derived one (0.063) reported by Song et al. (2021).

Our continental (0.070 ± 0.005) and oceanic (0.019 ± 0.002) mean DODs (see Table 1) are

substantially lower than those obtained in Voss and Evan (2020) (land: 0.1; ocean: 0.03). This
difference may be attributed to the different averaging approaches, which can have an important
impact on the calculations as it has been shown in Levy et al. (2009) (see their Figure 5). Based on
our method, we are giving the same "weight" at each grid cell (regardless of the amount of available
data in that grid cell throughout the study period) when we are calculating the domain (from regional
to global) average. Therefore, we are avoiding an overestimation of the spatial average since MIDAS
data availability is larger over/nearby deserts (see Figure 8-c in Gkikas et al. (2021)) where the higher
DODs are observed. To be more specific, when we are calculating the global long-term DOD average
based on the second branch (i.e., "Straight", the standard approach for the calculation of the average
value by considering all the available values in space and time) in Levy et al. (2009), we obtain a
climatological value equal to 0.047. Such different approaches for the calculation of the long-term
DOD averages might interpret and the deviations found between this study and Song et al. (2021).
Finally, the computed global mean MIDAS DOD is somewhat higher than those simulated by most
AeroCom Phase I models (Huneeus et al., 2011), being about 40% higher than the median (0.023);
nevertheless, it must be taken into account that most models account for the diurnal variation of DOD
in contrast to the single-measurements taken during MODIS overpass.

As expected, the interannual GLB DOD timeseries is driven by the variability in the NHE DOD

(Figure 10 ii-a) since the most widespread and intense dust sources are located in the Northern
Hemisphere. This is justified by their high temporal co-variation while a positive NHE-GLB offset is
constantly observed, being lower during boreal winter and autumn (up to 0.035) and maximum during
the high dust seasons (0.058). The fraction of monthly NHE AOD attributed to dust particles ranges
from 20% to 48% and the $R^2$ value between monthly AOD and DOD is equal to 0.94, both indicating
a dominant dust contribution. Over the study period (2003-2017), the NHE DOD yields a
climatological mean equal to 0.056 ± 0.004 (Table 1) ranging from 0.024 ± 0.015 (Dec 2005) to 0.121
± 0.050 (Mar 2012). In contrast, marine and biomass burning aerosols, rather than dust, regulate AOD
in the Southern Hemisphere (Figure 10 iii-a). SHE DODs are estimated to be low (0.008 ± 0.001),
with the maximum value (0.016 ± 0.016) recorded in February 2016. The contribution of dust aerosols
to the total aerosol load does not exceed 17% throughout the study period (Fig. 10 iii-a) and on
average it is equal to 8.2% ± 1.1%, which is in very good agreement with the findings by Kok et al.
(2021b).

A better view of the seasonal cycles of AOD, DOD and the DOD-to-AOD ratio can be obtained

by investigating their climatological patterns, representative for the period of interest (2003-2017).



On a global scale (Fig. 10 i-b), DODs peak between March and June (~0.045), and then decline until
November (0.018) before rising during boreal winter. Despite the monthly shifts between maximum
AOD and DOD averages, the seasonal cycles of the total aerosol and dust burdens are similar to a
large extent, whereas the contribution of mineral particles to the total extinction ranges from 16%
(November) to 33% (March-June). The MIDAS global DOD-to-AOD ratio (~23%) is close to the
values reported by Gelaro et al. (2017) and Kinne et al. (2006), ~22% and ~26%, respectively, but
higher than most of the model-derived estimations (12% - 28%) from the AeroCom Phase III (Gliss
et al., 2021). These discrepancies, excluding the aerosol parametrizations, may be partly due to the
different sampling between single-overpass satellite observations and reanalyses (Gelaro et al., 2017)
or models (Kinne et al., 2006) where the diurnal aerosol variability (Schepanski et al., 2009; Yu et
al., 2021) is included. In the NHE (Fig. 10 ii-b), the mean seasonal trend of DODs remains relatively
unchanged when compared with GLB; however, the hemispheric means (0.030-0.088) and the dust
fraction (24-41%) are higher. On the contrary, the weak signal of aeolian dust in SHE (Fig. 10 iii-b)
interprets the very low DODs (0.005 – 0.011) and their minor impact (6-12%) upon AOD magnitude.

The analysis presented above has also been conducted for each one of the 17 sub-regions

illustrated in Figure 7 in Gkikas et al. (2021), and the main findings are summarized in this paragraph.
Among the regional domains, a persistency of high DODs (>0.3), both at interannual and seasonal
scales, it is found only in BOD, which yields a long-term average value equal to $0.533 \pm 0.009$, being
almost double than WSA ($0.302 \pm 0.006$) and TAK ($0.246 \pm 0.020$) as illustrated in Table 1. However,
when focus is given to individual months, the maximum DODs over the study period (Fig. 11 vi-a)
and on their climatological levels are recorded in the Taklamakan Desert and can be as high as 0.868
(April 2007) and 0.600 (April), respectively. Comparable or even higher DODs than those computed
in BOD, are also evident for specific months in THA (Fig. 11 vii-a), GOG (Fig. 11 xii-a) and SSA
(Fig. 11 xv-a) as well as on the monthly timeseries (THA; Fig. 11 vii-b). Mineral particles'
contribution to the total AOD (i.e., blue curves in the seasonal cycle plots) is at least 50% over dust
sources or dust-abundant areas in N. Africa, Middle East and Asia and it is constantly higher than
70%, reaching up to 95%, in BOD (Fig. 11 i-b), WSA (Fig. 11 viii-b) and TAK (Fig. 11 vi-b). Over
downwind regions, such as EAS (Fig. 11 ix-b), GOG (Fig. 11 xii-b), MED (Fig. 11 xiii-b) and SSA
(Fig. 11 xv-b), the dust contribution can prevail over the non-dust portion (GOG, MED, SSA) while
in EAS does not exceed 30%, due to the predominance of anthropogenic aerosols. In the oceanic
areas of Tropical Atlantic and North Pacific, where large-scale dust transport is taking place, AOD
and DOD co-vary, indicating that the dust activity regulates the temporal variations of aerosols' load,
except during summer months in WNP (Fig. 11 xvi-a, xvi-b). Regarding the seasonal cycle of DOD,
the maximum values are recorded either during boreal spring (GOB, CAS, NME, SUS, TAK, EAS,





ENP, GOG, MED, WNP and SSA) or during boreal summer (THA, WSA, ETA, SME and WTA) or
are similar between the two high-dust seasons (BOD).

A final intercomparison of the MIDAS DODs against those derived by Ridley et al. (2016) and

Adebiyi et al. (2020), on a seasonal basis over the period 2004 - 2008, has been performed for 15
regions defined in Kok et al. (2021a) (see their Figure 2-b and Table 2). The obtained results are
illustrated in Figure 12. For the southern hemisphere regions (Figs. 12 –xiii, xiv, xv) as well as for
North America (Fig. 12-xii), MIDAS DODs are compared versus those from Adebiyi et al. (2020)
while for the remaining 11 domains (Figs. 12-i – xi) the results from Ridley et al. (2016) have been
utilized. As an overview, it is noted that the seasonal cycle among the three databases is commonly
reproduced, with a few exceptions (Mali-Niger, Kyzyl Kum, Southern Africa), whereas the DOD
uncertainties (represented by the error bars) are comparable. Regarding the magnitudes, MIDAS
DODs are mainly somewhat lower than those of Ridley et al. (2016) across the dust belt in contrast
to the outflow region of the Mid-Atlantic (Fig. 12-i). The obtained differences are mainly attributed
to the consideration of different models for accounting for the non-dust portion, the different
treatment of AODs (bias correction vs. quality filtering), the different versions of MODIS retrievals
(C006 vs C061), the consideration of multi-satellite observations instead of relying only on MODIS-
Aqua retrievals as well as to the different spatial scales (coarse vs. fine). In relative terms, the largest
deviations are found in the desert areas of the southern hemisphere where models struggle to represent
adequately the dust sources and the emitted amounts of mineral particles, thus affecting the dust
fraction ratio.

**5. Summary and conclusions**

The current study presents the first scientific exploitation of the MIDAS dataset (Gkikas et al.,

2021), which provides columnar mid-visible (550 nm) dust optical depth (DOD) at fine spatial
resolution (0.1° x 0.1°) and over a 15-year period (2003 – 2017). Taking advantage of the global
coverage of the MIDAS DOD product, we analyzed the contribution of dust aerosols to AOD at
various spatial and temporal scales. More specifically, we focused on 9 regions that account for the
majority of the global dust budget, including sources and downwind areas with the main dust transport
pathways. Such regions comprise the deserts extending across the "dust belt", North America,
Australia, South Africa and South America as well as maritime areas (Tropical Atlantic Ocean,
Mediterranean, North Pacific Ocean) receiving constantly large amounts of mineral particles from
the nearby deserts. At a further step, the interannual and intra-annual timeseries of DODs along with
their contribution to the total aerosol load (AOD), were investigated at global, hemispherical and
regional level.





According to our findings, the global long-term DOD average over the study period (2003-2017)
is equal to $0.032 \pm 0.003$, yielding a strong contrast between the contributions from the northern
$(0.056 \pm 0.004)$ and southern $(0.008 \pm 0.001)$ hemispheres. Our global estimations are almost identical
with those given by Ridley et al. (2016) and the CALIOP-derived estimate of Song et al. (2021), in
contrast to the MODIS-based given from the latter study. Nevertheless, when the global averages are
calculated separately over land $(0.070 \pm 0.005)$ and ocean $(0.019 \pm 0.002)$, our results differ
substantially than those found in Voss and Evan (2020), who reported continental and maritime DODs
equal to 0.100 and 0.030, respectively. The large deviations found with the latter study are attributed
to the different applied methodologies and averaging procedures followed. Moreover, we find very
good agreement, in terms of DOD magnitude and uncertainty, of the MIDAS seasonal DODs versus
those of Ridley et al. (2016) and Adebiyi et al. (2020) for 15 regions defined in Kok et al. (2021a).
Considering that the long-term DOD averages can be utilized for constraining global dust in climate
models, or can be used in several other applications, a detailed analysis is required for enlightening
the factors resulting in disagreements among studies. Likewise, our computed global DOD average
resides around the middle of the AeroCom (Huneeus et al., 2011) limits, being higher than the median
(0.023) and mean (0.028). However, in the model-based calculations the diurnal variability is taken
into account in contrast to the satellite-based estimations relying on single overpass measurements
per day.
Regarding the dust contribution to the total aerosol optical depth, the DOD-to-AOD ratio from
32% at N. Hemisphere drops down to 8% in S. Hemisphere while at global scale is about one quarter
(23%). The contradiction found between the two hemispheres, both for DOD and dust fraction, is
interpreted by the most pronounced dust activity recorded in the Bodélé Depression of the northern
Lake Chad Basin (DODs up to ~1.2), across the Sahel (DODs up to 0.8), in western parts of the
Sahara Desert (DODs up to 0.6), in the eastern parts of the Arabian Peninsula (DODs up to ~1), along
the Indus river basin (DODs up to 0.8) and in the Taklamakan Desert (DODs up to ~1). On the
contrary, the weaker emission mechanisms triggering dust mobilization over the spatially limited
sources of Patagonia, South Africa and interior arid areas of Australia do not favor the accumulation
of mineral particles at large amounts (DODs up to 0.4 at local hotspots), even during high-dust
seasons. Except for the Bodélé Depression, where the seasonal variability of the intense dust loads is
relatively weak, in the other dust sources of the N. Hemisphere, DODs exhibit a strong seasonal cycle
with maximum levels either during boreal spring or summer and minimum in boreal winter.
Over oceans, the main pathways of long-range dust transport are observed along the tropical
Atlantic and the northern Pacific, revealing a remarkable variation, within the course of the year, in
terms of intensity, latitudinal position and range. Saharan dust plumes, reaching the Caribbean Sea in
summer under the impact of the trade winds, are more abundant with respect to Asian dust, arriving





697 at the western coasts of the United States in spring under the impact of midlatitude cyclones. Due to

698 the convergence of the Shamal winds, blowing over the Arabian Peninsula, and the wind flow from

699 the subtropical anticyclone, dust aerosols originating in the Middle East can reach the western Indian

700 coasts in summer, crossing the Arabian Sea. Dust loads in the southern parts of the Red Sea are

701 maximized during boreal summer when Saharan or Middle East dust is transported, depending on the

702 zonal airflow. The intensity of dust burden in the Mediterranean forms a south-north gradient,

703 whereas a seasonal longitudinal shift of the maximum DODs, off the northern African coasts, is

704 evident attributed to the prevailing synoptic circulation.

705  The performed analysis here can serve as the basis of a follow-up study in which emphasis will

706 be given on DOD trends, from grid cell to global scale, in order to identify potential variations of

707 mineral atmospheric burden by exploring the temporal availability of the MIDAS dataset. It is

708 expected that thanks to the fine resolution of the MIDAS DOD it will be possible to investigate

709 alterations, throughout the time, of the emitted amounts at the sources and modifications of dust

710 transport patterns and subsequently assess the impact of the contributor mechanisms. Also, we have

711 provided a simple, yet flexible method (independent from other datasets) to calculate consistent

712 uncertainties across spatiotemporal scales, which will ease the use of the MIDAS dataset in future

713 studies (e.g. data assimilation).

714

**Acknowledgments**

715

716 Antonis Gkikas acknowledges support by the Hellenic Foundation for Research and Innovation

717 (H.F.R.I.) under the "2nd Call for H.F.R.I. Research Projects to support Post-Doctoral Researchers"

718 (Project Acronym: ATLANTAS, Project Number: 544). Vassilis Amiridis acknowledges support

719 from the European Research Council (grant no. 725698; D-TECT). Eleni Marinou was funded by a

720 DLR VO-Ryoung investigator group and the Deutscher Akademischer Austauschdienst (grant no.

721 57370121). Jasper F. Kok acknowledges support from National Science Foundation (NSF) grant

722 1552519. Carlos Pérez García-Pando acknowledges support from the European Research Council

723 (grant no. 773051; FRAGMENT), the AXA Research Fund, and the Spanish Ministry of Science,

724 Innovation and Universities (grant nos. RYC-2015-18690 and CGL2017- 88911-R). The authors

725 acknowledge support from the DustClim project as part of ERA4CS, an ERA-NET project initiated

726 by JPI Climate and funded by FORMAS (SE), DLR (DE), BMWFW (AT), IFD (DK), MINECO

727 (ES), and ANR (FR), with cofunding by the European Union (grant no. 690462). PRACE (Partnership

728 for Advanced Computing in Europe) and RES (Red Española de Supercomputación) are

729 acknowledged for awarding access to the MareNostrum Supercomputer in the Barcelona

730 Supercomputing Center. We acknowledge support of this work by the PANhellenic infrastructure for

731 Atmospheric Composition and climatE chAnge (PANACEA) project (grant no. MIS 5021516),



which is implemented under the Horizon 2020 Action of "Reinforcement of the Research and
Innovation    Infrastructure",    funded    by    the    Operational    Programme    Competitiveness,
Entrepreneurship, and Innovation (NSRF 2014–2020) and cofinanced by Greece and the European
Union (under the European Regional Development Fund). NOA members acknowledge support from
the Stavros Niarchos Foundation (SNF). The authors acknowledge support by the COST Action
"InDust" (grant no. CA16202), supported by COST (European Cooperation in Science and
Technology). The authors would like to thank Andrew Mark Sayer for his valuable and constructive
comments. The authors would like also to thank Thanasis Georgiou for developing the ftp server on
which the MIDAS data set is stored.

**Data availability**

The MIDAS dataset has been developed in the framework of the DUST-GLASS project (grant no.
749461; European Union's Horizon 2020 Research and Innovation programme under the Marie
Skłodowska-Curie Actions) and it is available at: https://doi.org/10.5281/zenodo.4244106.

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





**Table 1:** Annual and seasonal DOD averages, representative for the period 2003-2017, along with the associated
uncertainty. The first three rows refer to the whole globe (GLB), the global land (GLB-land) and global ocean (GLB-
ocean). In the fourth and fifth line are given the results for N. Hemisphere (NHE) and S. Hemisphere (SHE) DODs
whereas in the rest 17 entries the corresponding results for selected subregions (denoted with colored rectangles in Fig. 7
in Gkikas et al. (2021)) are given.

| REGION | ANNUAL | DJF | MAM | JJA | SON |
|---|---|---|---|---|---|
| **GLB** | 0.032 ± 0.003 | 0.025 ± 0.004 | 0.043 ± 0.005 | 0.040 ± 0.005 | 0.022 ± 0.004 |
| **GLB-land** | 0.070 ± 0.005 | 0.063 ± 0.008 | 0.104 ± 0.011 | 0.083 ± 0.010 | 0.049 ± 0.007 |
| **GLA-ocean** | 0.019 ± 0.002 | 0.015 ± 0.003 | 0.026 ± 0.003 | 0.023 ± 0.003 | 0.012 ± 0.003 |
| **NHE** | 0.056 ± 0.004 | 0.043 ± 0.005 | 0.085 ± 0.009 | 0.071 ± 0.008 | 0.036 ± 0.005 |
| **SHE** | 0.008 ± 0.001 | 0.010 ± 0.003 | 0.008 ± 0.002 | 0.006 ± 0.002 | 0.008 ± 0.003 |
| **BOD** | 0.533 ± 0.009 | 0.483 ± 0.018 | 0.614 ± 0.020 | 0.603 ± 0.017 | 0.451 ± 0.013 |
| **GOB** | 0.092 ± 0.007 | 0.074 ± 0.010 | 0.189 ± 0.023 | 0.078 ± 0.010 | 0.056 ± 0.005 |
| **CAS** | 0.126 ± 0.007 | 0.084 ± 0.012 | 0.158 ± 0.016 | 0.144 ± 0.011 | 0.100 ± 0.007 |
| **NME** | 0.227 ± 0.006 | 0.120 ± 0.009 | 0.319 ± 0.016 | 0.271 ± 0.011 | 0.186 ± 0.009 |
| **SUS** | 0.018 ± 0.001 | 0.009 ± 0.002 | 0.033 ± 0.005 | 0.021 ± 0.003 | 0.010 ± 0.001 |
| **TAK** | 0.246 ± 0.020 | 0.114 ± 0.015 | 0.504 ± 0.047 | 0.259 ± 0.030 | 0.130 ± 0.018 |
| **THA** | 0.198 ± 0.007 | 0.086 ± 0.006 | 0.291 ± 0.013 | 0.424 ± 0.033 | 0.109 ± 0.006 |
| **WSA** | 0.302 ± 0.006 | 0.199 ± 0.008 | 0.362 ± 0.015 | 0.418 ± 0.016 | 0.237 ± 0.009 |
| **EAS** | 0.077 ± 0.005 | 0.072 ± 0.014 | 0.130 ± 0.012 | 0.056 ± 0.010 | 0.048 ± 0.006 |
| **ENP** | 0.020 ± 0.002 | 0.011 ± 0.002 | 0.047 ± 0.005 | 0.017 ± 0.004 | 0.013 ± 0.002 |
| **ETA** | 0.146 ± 0.007 | 0.109 ± 0.011 | 0.169 ± 0.015 | 0.202 ± 0.015 | 0.093 ± 0.009 |
| **GOG** | 0.309 ± 0.021 | 0.417 ± 0.032 | 0.416 ± 0.066 | 0.064 ± 0.021 | 0.100 ± 0.022 |
| **MED** | 0.081 ± 0.003 | 0.052 ± 0.008 | 0.106 ± 0.009 | 0.096 ± 0.006 | 0.066 ± 0.005 |
| **SME** | 0.250 ± 0.008 | 0.154 ± 0.009 | 0.318 ± 0.016 | 0.394 ± 0.020 | 0.166 ± 0.008 |
| **SSA** | 0.326 ± 0.013 | 0.309 ± 0.015 | 0.494 ± 0.041 | 0.241 ± 0.054 | 0.199 ± 0.020 |
| **WNP** | 0.028 ± 0.002 | 0.017 ± 0.003 | 0.064 ± 0.008 | 0.023 ± 0.006 | 0.018 ± 0.002 |
| **WTA** | 0.035 ± 0.003 | 0.006 ± 0.002 | 0.035 ± 0.005 | 0.090 ± 0.009 | 0.017 ± 0.004 |












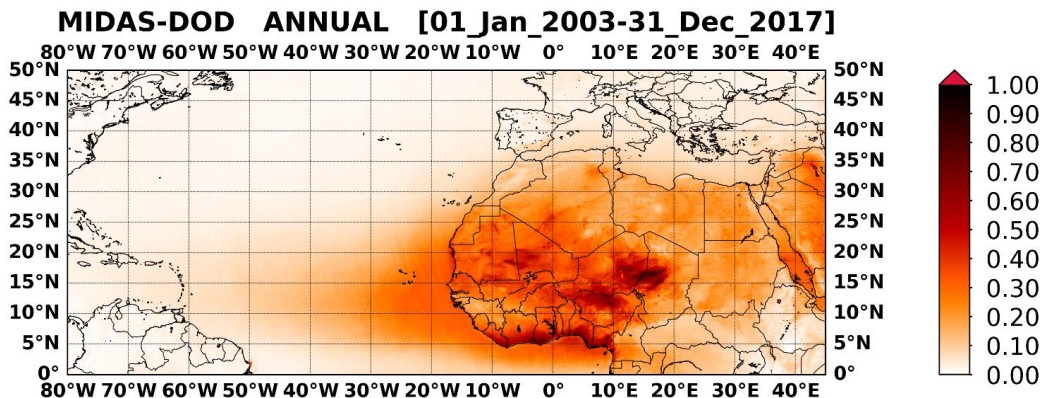

**Figure 1:** Geographical distribution of the MIDAS annual DOD at 550nm, representative for the period 1 January 2003 – 31 December 2017, over North Africa, the Tropical Atlantic Ocean and the broader Mediterranean basin.

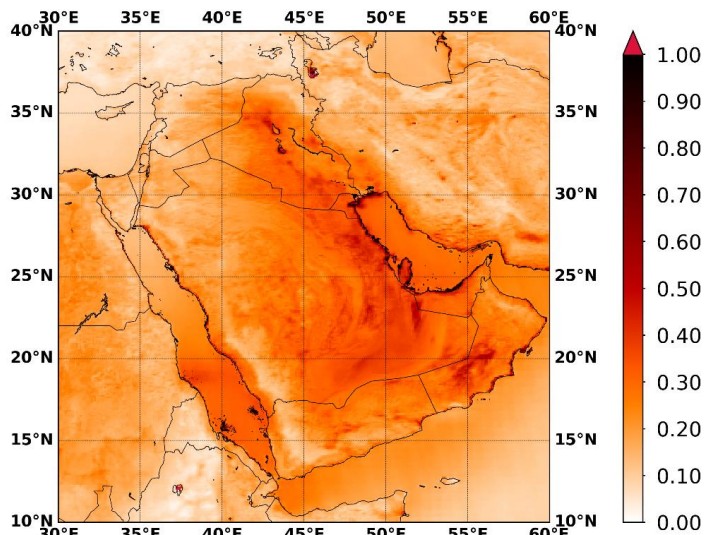

**Figure 2:** As in Figure 1 but for the broader area of the Middle East.



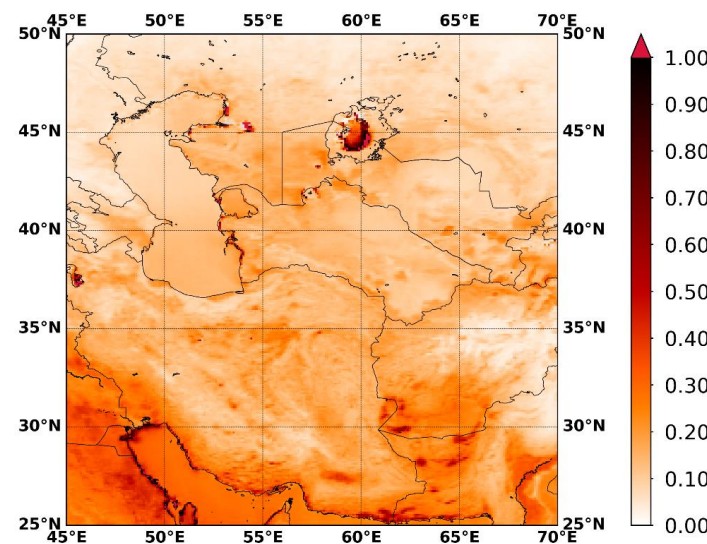

**Figure 3:** As in Figure 1 but for central and southwestern Asia.

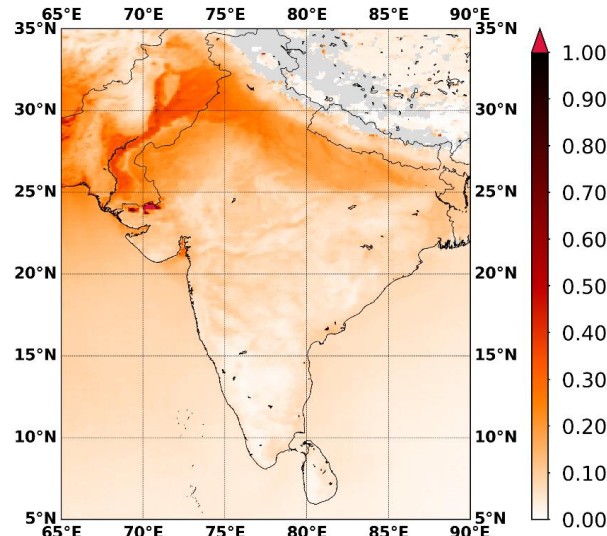

**Figure 4:** As in Figure 1 but for the Indian subcontinent.





**MIDAS-DOD ANNUAL [01_Jan_2003-31_Dec_2017]**

**Figure 5:** As in Figure 1 but for East Asia and the North Pacific Ocean.

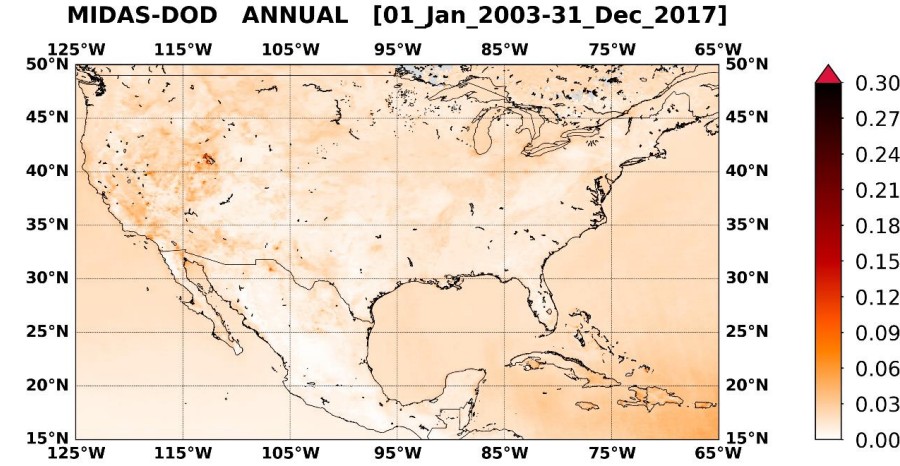

**Figure 6:** As in Figure 1 but for North America.

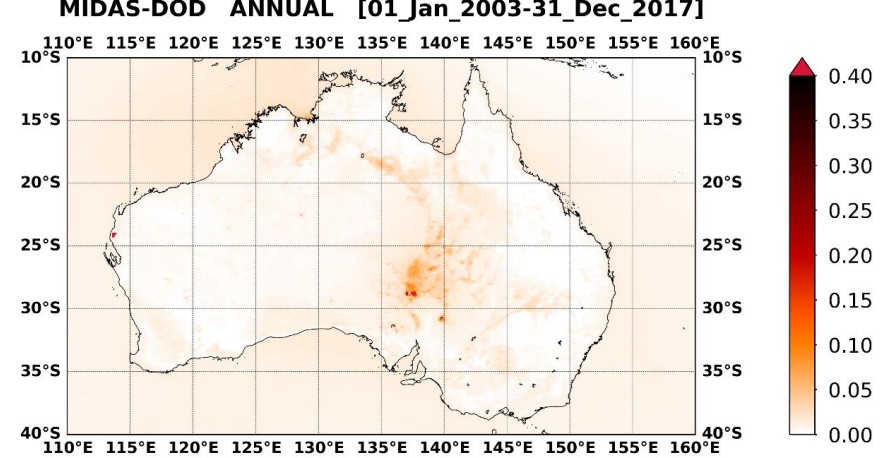

**Figure 7:** As in Figure 1 but for Australia.




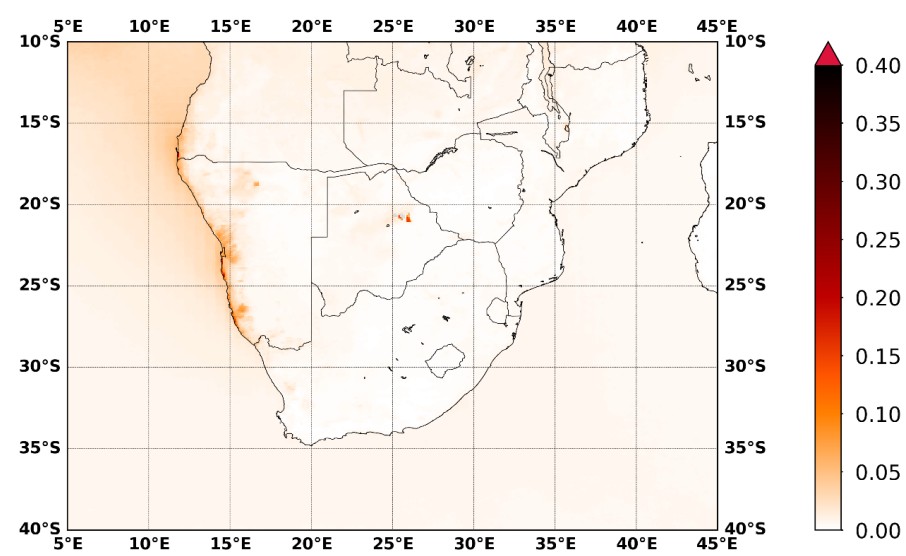

**Figure 8:** As in Figure 1 but for Southern Africa.

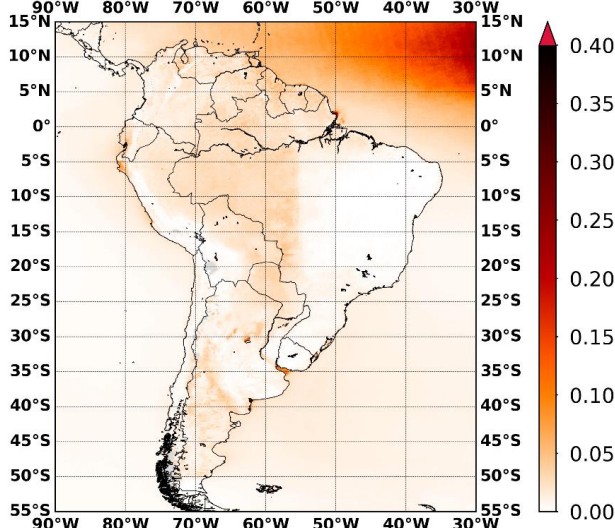

**Figure 9:** As in Figure 1 but for South America.



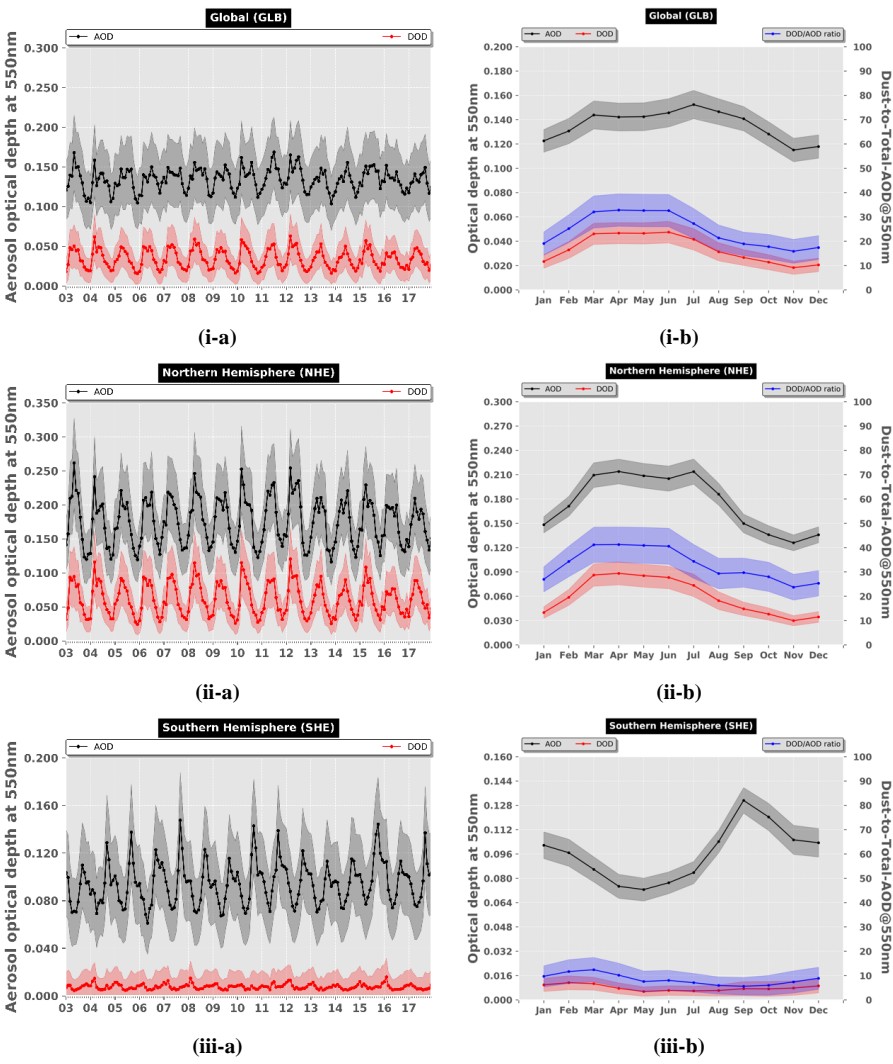

**Figure 10:** Inter-annual **(-a)** and intra-annual **(-b)** variability, representative for the period 2007 – 2016, of monthly MODIS AOD$_{550nm}$ (black curve) and DOD$_{550nm}$ (red curve) regionally averaged for: **(i)** the whole globe (GLB), **(ii)** the Northern Hemisphere (NHE) and **(iii)** the Southern Hemisphere (SHE). The blue curves in the intra-annual plots depict the dust-to-total AOD$_{550nm}$ ratio (expressed in percentage; right y-axis). The shaded areas correspond to the total uncertainty.





**(i-a)**                    **(i-b)**

**(ii-a)**                   **(ii-b)**

**(iii-a)**                  **(iii-b)**

**(iv-a)**                   **(iv-b)**









**(ix-a)**

**(ix-b)**

**(x-a)**

**(x-b)**

**(xi-a)**

**(xi-b)**

**(xii-a)**

**(xii-b)**





(xiii-a)        (xiii-b)

(xiv-a)        (xiv-b)

(xv-a)        (xv-b)

(xvi-a)        (xvi-b)





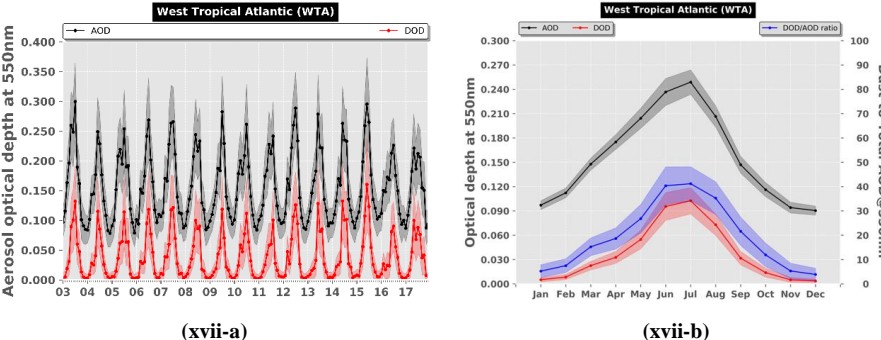

(xvii-a)                                      (xvii-b)

**Figure 11:** Inter-annual **(-a)** and intra-annual **(-b)** variability, representative for the period 2003 – 2017, of monthly MODIS $AOD_{550nm}$ (black curve) and $DOD_{550nm}$ (red curve) regionally averaged for: **(i)** Bodélé Depression (BOD), **(ii)** Gobi Desert (GOB), **(iii)** Central Asia (CAS), **(iv)** North Middle East (NME), **(v)** southwest United States (SUS), **(vi)** Taklamakan Desert (TAK), **(vii)** Thar Desert (THA), **(viii)** West Sahara (WSA), **(ix)** East Asia (EAS), **(x)** East North Pacific (ENP), **(xi)** East Tropical Atlantic (ETA), **(xii)** Gulf of Guinea (GOG), **(xiii)** Mediterranean (MED), **(xiv)** South Middle East (SME), **(xv)** Sub-Sahel (SSA), **(xvi)** West North Pacific (WNP) and **(xvii)** West Tropical Atlantic (WTA). The shaded areas in the inter and intra-annual plots correspond to the total uncertainty. The blue curves in the intra-annual plots represent the percentage contribution of dust optical depth (DOD) to the aerosol optical depth (AOD).





**Figure 12:** Seasonal DODs, representative for the period 2004 – 2008, based on the MIDAS dataset (orange bars), Ridley et al. (2016) (blue bars) and Adebiyi et al. (2020) (blue bars), for 15 regions (their names are given at the top of each plot) defined in Kok et al. (2021a) (see Table 2). The error bars represent the estimated uncertainties. From i to xi, the blue bars correspond to the Ridley et al. (2016) results whereas for the remaining regions MIDAS DODs are compared against the corresponding levels obtained by Adebiyi et al. (2020).