# Peer review of "Quantification of the dust optical depth across spatiotemporal scales with the"

_Atmospheric Chemistry and Physics, 2021_

## Author Comment (AC1)

We would like to thank the Reviewer for his/her report. Below are given our point-by-point replies (regular font) to each comment (bold font) raised by the Reviewer.

**This manuscript presents a new but previously published data set of global, daily dust optical depth (DOD) at 0.1Ë spatial resolution and the resultant spatio-temporal features of dust activity over global major dust hotspots. While this is an interesting data set with a lot of potential scientific applications, and this article is well organized and well written, it looks more like a review article rather than an original research article. Indeed, this article reminds me of Ginoux et al (2012) published in Review of Geophysics, which reviewed global dust sources and their seasonal features based on a global, daily, 0.1 data set of DOD derived from MODIS measurements. My main suggestion for the current authors is thereby to decide if you want it to be a review article or a research article.**

In the submitted manuscript, we clearly stated that the goal of our study is to describe the regime of dust aerosols over specific regions of the planet hosting major dust sources, as well as those undergoing dust transport. We also focus on analyzing the monthly and year-to-year variability between 2003 and 2017 at global and regional scales. Actually, we are not summarizing all published works on dust optical depth (DOD) for each region of interest (ROI), what would be the case of a review article. Of course, reference and comparisons are made to previous studies, but this is done for justifying the validity of the MIDAS DODs, thus expanding the already comprehensive assessment analysis presented in the first paper (Gkikas et al., 2021), in order to support our findings and/or to assess the current state of the art on existing dust related datasets.

We agree with the reviewer that the structure of our manuscript reminds that of Ginoux et al. (2012) but the contents of the two papers, to our point of view are different. Here we are focusing on DOD in contrast to Ginoux et al. (2012), who is identifying natural and anthropogenic dust sources based on the Frequency of Occurrence (FoO) of exceedances above a defined DOD threshold. It is true that Section 4, in Ginoux et al. (2012), presented some results about their MODIS-derived DOD, but most of their discussion is focusing on the comparison against AERONET retrievals and also a short summary of AOD and DOD patterns is provided at global scale, based on their seasonal spatial distributions depicted in Figure 4. Even though these findings can be considered similar with those obtained here, there are also distinct differences between the two studies, consisting in the different applied methodologies for the derivation of DOD, the spatial coverage of the two datasets as well as the length of their time series (Ginoux stops in 2009). In terms of differences with the study by Ginoux et al. (2012), we would like to remind that, as described in Gkikas et al. (2021), the MIDAS DOD is derived via the synergistic implementation of quality-assured MODIS-Aqua AOD and the MERRA-

2 fraction of AOD that is due to dust (MDF). On the contrary, in Ginoux et al. (2012), the DOD is associated with MODIS AODs when the Angstrom exponent (size optical property) and the single scattering albedo (nature optical property), both obtained from MODIS, are lower than defined upper thresholds. Therefore, apart from the differences in the applied methodologies there is also a difference in the terminology of how dust optical depth is defined. Here, we are extracting the contribution on non-dust aerosol species by utilizing MDF (with all the uncertainties as these have been discussed thoroughly in Gkikas et al. (2021)) while in Ginoux et al. (2012) the DOD resembles more dust dominant conditions (with all the uncertainties of MODIS size and nature optical properties).

It is also worth mentioning that between the two DOD products there is also a difference in their spatial coverage. In Ginoux et al. (2012), the DOD is available only above land either because their focus was on the identification of dust sources or on the utilization of the single scattering albedo (retrieved only over land by the MODIS Deep Blue algorithm) for screening non-dust AODs. In our study, DOD is provided above sources, nearby downwind regions and oceans. Likewise, we would like to point out that our dataset spans from 2003 to 2017 (15 years) in contrast to Ginoux et al. (2012) which covers a shorter time period (2003-2009). Furthemore, in the current work we are processing the latest version of MODIS retrievals (i.e., Collection 6.1) in contrast to Ginoux et al. (2012), who analyzed the same MODIS data obtained, however, by a prior version (i.e., Collection 5.1) of the retrieval algorithm. Finally, we would like to highlight that our study also quantifies and analyzes the average AOD and DOD along with their monthly and interannual variability and associated uncertainties at global, hemispherical and regional scales, which is not done by Ginoux et al.; (2012). For the sake of clarity, we clarify that of course, our intention is not to criticize the work of Ginoux et al. (2012), neither in our replies nor in the manuscript, but just to answer the specific comment proving that the two papers and data sets are of a different nature.

**Whether you are transferring or not, I suggest making clear about the quality of your data set throughout the paper. For example when you show the annual mean DOD over certain region, please mark the questionable regions, such as Gulf of Guinea.**

Some notifications (in lines 225-228, 335-336, 705-718 of the revised version of the manuscript) to the readers and potential users of the dataset were made in the revised manuscript.

**Talking about data quality, does MERRA2 typically assign low MDF over agricultural dust source regions? I didn't find a formal assessment of MDF over agricultural areas, such as the Great Plains, southeastern Australia, either from the current manuscript or Gkikas et al (2021).**

In Section 4.1 in Gkikas et al. (2021), we presented the evaluation of MDF versus the corresponding dust fraction derived from the LIVAS dataset. To perform this, we reproduced the annual (Figure 1) and seasonal (Figure S2) global maps, representative for the entire study period, along with the corresponding spatial distributions of some primary evaluation metrics (Figure 2; Figure S2; Figure S3 and Figure S4). Based on these plots, we think that we provided sufficient information covering all aspects, necessary for a complete evaluation of MDF. Focusing on the regions mentioned by the reviewer, we can see that in the Great Plains and southeastern Australia, the annual MDF is mainly lower than 30% while LIVAS dust fraction can reach up to 40% (Figure 1). Seasonal maps (Figure S2) provide an insight of how dust fraction variation, within the course of the year, is represented by MERRA-2 and LIVAS. Hence, it is evident that there is an underestimation of MDF in this case, whereas the correlation coefficient between LIVAS and MIDAS reaches up to 0.6. This "deficiency" can be explained either by the fact that dust sources in MERRA-2 are based on Ginoux et al. (2001), accounting mostly for natural dust emission areas, or by the LIVAS temporal availability (Figure S3-i) and grid-cell representativeness (Figure S3-ii), as it has been discussed in Gkikas et al. (2021).

**Another general question about the dataset is that if we grid this fine-scale product to the original MERRA2 grid, do we get pretty much MERRA2 DOD? Since MERRA2 assimilates MODIS AOD and MDF is defined as the ratio between MERRA2 DOD and MERRA2 AOD.**

In Section 4.3 of Gkikas et al. (2021), we are comparing the DODs derived by MIDAS, LIVAS and MERRA-2. MIDAS and MERRA-2 data have been regridded to 1° x 1° spatial resolution to match those of LIVAS while the study period (2007 - 2015) is restricted by the LIVAS availability. Based on our performed analysis a comparison between MIDAS and MERRA-2 DODs at global (Figure 6-a), hemispherical (Figures 6-b, 6-c) and regional (Figure S7) level, is shown. For brevity reasons, we would like to avoid repeating here our discussion presented in Gkikas et al. (2021).

Regarding the last part of the reviewer's comment, we would like to clarify some points in order to explain the declinations between MERRA-2 and MIDAS DODs. MERRA-2 assimilates bias-corrected, neural-network retrieved (and AERONET-calibrated) AOD that is derived over land from the Dark Target algorithm (Collection 5 instead of Collection 6.1 used here) radiances, thus excluding bright surfaces over which MISR retrievals are assimilated (without any bias correction). Additional aerosol observations which are considered in the MERRA-2 assimilation scheme are those from AVHRR (over ocean only) and AERONET. The increments induced by the assimilated total AODs are distributed among the five aerosol species simulated in MERRA-2 using model background information. Therefore, DOD is determined by the underlying GOCART module, the impact of the

GEOS-5 meteorological driver, their two-way feedbacks and the assimilation-driven adjustments. On the other hand, in MIDAS, the columnar dust optical depth is determined by MODIS AOD (which is quite reliable as it has been shown in numerous evaluation studies) and MERRA-2 MDF which is considerably reliable in areas where dust presence is evident (see Section 4.1 in Gkikas et al., (2021)).

**If you decide to retain the current article as a research article, it is necessary to define a clear scientific question. I do not see the scientific question or hypotheses in the current manuscript.**

Our purpose (as stated in lines 107-108, 651-657 of the revised manuscript) is to describe the regime of dust aerosols, at global and regional scale, relying on the MIDAS dataset, which was introduced and evaluated in the AMT published methodological paper by Gkikas et al., 2021, and to understand differences among different existing DOD databases.

**If you decide to transfer to review article, I would suggest to highlight what are the new findings since Ginoux et al (2012). Maybe one aspect is about the long-range transported dust, since the data set Ginoux et al (2012) analyzed only covered land.**

As explained in our reply to the first comment, which stated the differences between the two studies, our aim is a scientific and not a review article.

**I'll be happy to review this paper again, either in its review article form or a more scientific question-driven form.**

We would like to thank the reviewer for his/her effort and the willingness to review the new/revised document.

---

## Author Comment (AC2)

We would like to thank the reviewer for his/her thorough report. We can understand the arguments raised by the reviewer related to the inherent limitations of MODIS retrievals and MERRA-2 products, used for the development of the MIDAS DOD dataset (Gkikas et al., 2021), and this is why we have added some further clarifications (and considerations for any potential user of the MIDAS dataset) in the revised manuscript. Please find below our point-by-point replies (regular font) to each comment (bold font) raised by the Reviewer.

**Summary and Recommendation**

**This paper presents a quantitative estimation of spatial and temporal variability of dust globally. It uses a data set that coalesces satellite (MODIS total aerosol optical depths) and aerosol model outputs (MERRA-2 reanalysis) to provide a +15-year global dust aerosol optical depth (DAOD) at 0.1x0.1 degree resolution. The dataset (called MIDAS) was introduced in a separate study (Gkikas et al, 2021a) and this paper is an application of the MIDAS dataset. This study consists in a statistical analysis of the average spatial distribution of DAOD over each continent and discusses how the results compare with previous studies. In addition, monthly time series and inter-annual variability plots are presented for representative major dust sources.**

**Overall, this is an impressive amount of very detailed work and provides an overall picture of dust distribution with excellent graphics. In addition, it is a good idea to create a gridded high spatial resolution dataset.**

We would like to thank the reviewer for acknowledging our effort to provide a detailed description of mineral particles' load in conjunction with high-quality graphics.

**However, I have one critical point in this analysis and in my opinion, it is disqualifying for publication as it is. I think that this analysis provides a global picture from data sources that are not suitable for being merged in this way. I think the paper could eventually be published but significant modifications should be added, and concerns addressed.**

Despite the fact that this comment is questioning the Gkikas et al. (2021) already published and highlighted methodological paper in AMT, we have tried to answer the specific reviewers' arguments in the comments section below.

**More detailed Comments**

**As a matter of disclosure, I work in one of the satellite algorithm development teams used in this study.**

**I recognize the effort in trying to create a more complete global picture of dust distribution. While it is true that tremendous advances have been made in global aerosol detection and concentration observation (global AOD), there hasn't been significant progress in observational global aerosol type identification from satellites. The MODIS sensors are not sensitive enough to do aerosol type identification for aerosol loadings below the range AOD=0.15-0.2 (and this can be debatable too because it highly depends on the surface). The corresponding algorithms are designed to mitigate but not eliminate this problem. So, there is a clear observational under sampling of dust in medium to low concentrations. The advances in aerosol modeling in the last decade, particularly in transport of air masses that correctly place the location and arrival time of any air mass anywhere of the world to the point that it can be used in air pollution forecasting. However, aerosol transport modeling still has significant difficulties in the generation and characterization of aerosol sources and specifically, dust.**

**Therefore, my concern is the nature of the MIDAS dataset. It is a mixture of observational and model data. It offers a tremendously practical dataset. However, it casts a doubt on the reliability of the data. The satellite source data is still quite imperfect to be merged this way. Specifically, there are 3 MODIS aerosol algorithms used in this study: Dark Target- Land, Dark-Target-Ocean and Deep Blue (only over land). All of them have mutual inconsistencies in aerosol detection which are particularly manifest in low-to-medium aerosol loadings. For example, in the Sahel region in Africa, both algorithms disagree depending on the surface assumed by each of them and on the aerosol loading in the scene.**

It is true that the three MODIS retrieval algorithms are merged based on Sayer et al. (2014) in order to create a unified AOD product. Due to their different assumptions, there are inconsistencies which can be evident between land and sea surfaces. Over land, the selection of the best retrieval, between Dark Target and Deep Blue, relies on the internal QA checks and the surface reflectance (i.e., NDVI), as it has been described in Sayer et al. (2014). As expected, these inherent limitations can be transferred to MIDAS DOD. Nevertheless, all these points have been discussed in Gkikas et al. (2021). For instance, regarding the Sahel, we have mentioned some potential reasons for interpreting the evaluation results of MIDAS DOD against AERONET observations and its intercomparison against LIVAS and MERRA-2 DODs (see Section 4.3.1 in Gkikas et al. (2021)). Likewise, there is a brief discussion in the submitted manuscript (see Lines 217 - 225). We are aware of the drawbacks

of our approach but we have included an uncertainty analysis in the previous paper based on published MODIS related uncertainties.

It is also important to note that recent papers by Kok and colleagues (Kok et al. 2021a, 2021b) provide clear evidence of the usefulness of such datasets, despite their inherent uncertainties. These papers used inversion modeling to integrate an ensemble of global model simulations with observational constraints on the dust size distribution, extinction efficiency, and regional DOD. The latter was based on the dataset presented in Ridley et al. (2016), which combined MODIS AOD, dust fraction from models and AERONET. The comparison of the inverse model against independent measurements of dust surface concentration and deposition flux showed that the inverse modeling dataset was more accurate than current model simulations and the MERRA-2 dust reanalysis product. Our dataset is conceptually similar to that of Ridley et al. (2016) and broadly consistent at global annual and regional seasonal scales as we show in Section 4.2 of the manuscript. The added value of our dataset is, among other issues, its availability for a long period and across spatiotemporal scales. In addition, our paper provides and uses a simple, yet flexible method (described in Section 3) to estimate consistent uncertainties across spatiotemporal scales, which we believe will ease the use of the MIDAS dataset in future studies.

**The methodology relies on the assumption of using MERRA-2 DAOD/AOD ratios to provide the proportion of dust present in the pixel observed by the satellite. This assumes that the model is correct in not only placing dust in the selected pixel but also the proportion of dust is correct. However, it is well documented that global dust models are still having serious discrepancies not only in quantitative terms in aerosol loading but also in activating dust sources. Several studies have pointed this out. (Pu and Ginoux, 2015; Wu et al, 2018,2019,2020; Gliß et al, 2021). While these studies did not specifically address MERRA-2, they do highlight that global dust models still are struggling to consistently produce realistic dust simulations and it is an evolving topic. Certainly, there are encouraging advances such as those from Kok's group in UCLA but the modeling of quantitative dust generation is still a subject in progress. While MERRA-2 partially overcomes this weakness by assimilating total AODs from MODIS, it still has limitations regarding the generation of different aerosol types including dust. With all these mutual discrepancies in dust generation, how could a user trust that outputs provided by MERRA-2 is any better than the other models?**

The issues hampering the ability of the current state-of-the-art atmospheric-dust models to accurately simulate dust fields are well documented in literature. For brevity reasons, we are not going to list all of them since it is out of the scope of the current study. We agree with the reviewer in that the

assimilation of AODs "helps" MERRA-2 to partially overcome these deficiencies but unfortunately there are remaining underlying weaknesses (i.e., misrepresentation of dust sources). For this reason, in Gkikas et al. (2021), we performed an exhaustive evaluation of MDF with respect to CALIPSO/LIVAS dust fraction (see Section 4.1) just to ensure the validity of the former parameter, which is quite critical for the derivation of the MIDAS DOD. The question raised by the reviewer is interesting but in order to provide a sufficient answer we have to evaluate in parallel model and reanalysis dust-relevant products against reference observations. We are expecting a better performance by MERRA-2 since meteorological and aerosol observations are jointly assimilated while aerosol–radiation interactions and the two-way feedbacks with atmospheric processes are taken into account (Gelaro et al., 2017). We also re-emphasize that our product includes an uncertainty estimate at pixel and daily level as described in Gkikas et al. (2021).

**In the paper and in Gkikas et al, 2021a, I noted behaviors that are difficult to interpret, and it is not clear whether it is sourced to the satellite or model data because the data was mixed at its generation. For example, in Gkikas et al, 2021a the database puts dust in places where no dust has been observed (Arctic, upper right corner in figure 8a) and it includes pixels that only have less than 20-30 observations in 15 years of daily sampling. It does not include dust that is not visible from space nor modeled by MERRA-2 (such as high latitude dust such as Alaska, Iceland and Greenland that occur mostly in cloud conditions and their sources are not included in MERRA=2). There are inconsistencies in the data that makes it look unphysical for example, in figure 1 the sharp discontinuity in aerosols along the coast of the Gulf of Guinea (along 5 degrees N) where AODs are high in the inland side but drop sharply in the immediate ocean. Aerosols do not behave like that (most likely this is an issue due the land and ocean satellite algorithms). These inconsistencies are present in the original data used, that is in the daily Level 2 satellite data. For example, this image (see https://go.nasa.gov/2VaLKLv and https://go.nasa.gov/3ihzpxW) displays markedly different AODs between ocean and land in an air mass (containing smoke and dust in each case) advecting from land to ocean. Perusing other days in the same webpage will show a similar pattern. Clearly aerosol concentrations cannot change so drastically in such short distance, and this is clearly an artifact. Also, figure 9 has a sharp straight line North-South in the center of Brazil: aerosols do not behave like that.**

We would like to thank the reviewer for this comment helping us to better explain the MIDAS dataset and clarify its weak points. Regarding the few extremely high DODs in the Arctic, seen on the annual map of Figure 8 in Gkikas et al. (2021), these came up from isolated values (we have checked all the daily DOD global maps) recorded in MAM (Figure 9-ii in our previous paper), in combination with very low data availability for the area. This small availability (less than 1%) of MIDAS data in the

aforementioned region is indicated in Figure 8-c. Moreover, the obtained DODs are associated with maximized uncertainties as illustrated in Figure 8-b. Therefore, they are not reliable and can be considered as artifacts. This is the reason why we are presenting along with the climatological DOD maps the corresponding spatial distributions of the MIDAS temporal availability and uncertainty. In the Gulf of Guinea, the transition of DOD levels from land to sea is not smooth and this "strange" behavior is introduced by the raw MODIS AODs, as correctly stated by the reviewer (see also the above discussion in a similar previous comment). This "artifact", which cannot be easily bypassed in the MIDAS DOD product, has been discussed in Gkikas et al. (2021) as well as in the submitted manuscript. Finally, about the MIDAS deficiencies in S. America we believe that our explanation is straightforward and answers this comment. We are copying below the relevant part from the text (see lines 529 - 536 in the submitted manuscript).

*"Finally, the latitudinal zone of weak DODs in the western parts of Brazil, fading down abruptly eastwards of ~58° W, indicates an artifact of the MIDAS product that becomes more evident in SON (Fig. S9-iv). This peculiar pattern is induced by the MERRA-2 dust fraction (results not shown here) which is used for the derivation of MIDAS DOD from the MODIS AOD. An additional deficiency is the relatively large DODs over an area where biomass burning particles, emitted at enormous amounts by extended wildfires, clearly dominate over other aerosol species. Under these conditions, the non-dust AODs are very high as well as their relevant uncertainties (Eqs. 5-7 in Gkikas et al. (2021)) while the reliability of the MERRA-2 dust fraction downgrades there (see Fig. 2 in Gkikas et al. (2021))."*

**So, what casts a doubt to this study is how one can reliably trust what MIDAS is showing? Specifically, if it shows a specific feature or trend that do not agree with independent sources of data or observations, is this discrepancy sourced in the observational data or in the model? How a user would be able to understand and trace the source of discrepancy? Would the user be able to conclude that it is an actual geophysical feature?**

We think that our replies in the previous comments as well as the discussion in both papers provide the necessary transparency towards the reader on the uncertainty and limitations of our dataset. Regarding the comparisons against independent sources, in Gkikas et al. (2021) we have performed an evaluation of the MIDAS DOD product versus the dust optical depth, as this has been defined based on AERONET retrievals (see Sections 2.4 and 4.1). Focusing on the Sahel, in Gkikas et al. (2021) it is stated that:

*"Across the Sahel, maximum root mean square error (RMSE) levels (up to 0.26) are recorded (Fig. 4c) due to the intense loads and strong variability in the Saharan dust plumes. Regarding biases,*

*positive deviations of up to 0.08 are computed in most AERONET sites in the area, while the largest negative offsets (down to −0.14) are recorded at the stations of Ilorin and Djougou (near to the coasts of the Gulf of Guinea), in agreement with Wei et al. (2019b). Several reasons may explain the obtained MIDAS–AERONET differences over the above-mentioned stations, taking into account that the MDF is generally well reproduced. The first one is related to the MODIS retrieval algorithm itself and, more specifically, to the applied aerosol models, surface reflectance, and cloud-screening procedures (Sayer et al., 2013). The second factor is the omission of fine DOD in AERONET data, which would likely reduce the positive biases. However, its contribution to the total dust AOD is difficult and probably impossible to be accurately quantified."*

We also have to point out that any of the users of this dataset has to take into account the uncertainty estimation that "accompanies" the data as presented in Gkikas et al., 2021.

**As far as the scientific questions addressed by this paper, I think they are good ideas to address.**

**However, they can be (partially) addressed with similar satellite only databases. For example, this paper presents an analysis of the major dust producing basins, places we already know are the major dust sources and in general they are cloudless which means satellite coverage is very good. So, the addition of model data does not contribute meaningful additional information as far as understanding of these sources which were already addressed in other recent satellite studies (for example, see Voss and Evan, 2021 and Gupta et al, 2020).**

In the submitted manuscript we are presenting dust aerosols' regime not only above dust basins and sources but we are also including downwind areas where mineral particles can mixed with other aerosol species (e.g., Mediterranean, Tropical Atlantic Ocean). Therefore, over these regions, it is necessary an approach for discriminating dust from non-dust aerosols. In MIDAS, we are using MDF (MERRA-2 reanalysis products) instead of setting cut-off/upper thresholds on size/nature relevant optical properties. It is just a different approach and definitely we are not claiming that our methodology is the best one or that there are no artifacts. We think that we are pretty clear with what are the advantages and limitations of the MIDAS dataset. Moreover, in both papers we have highlighted the innovative elements of our dataset with respect to the previous ones and how MIDAS can complement existing gaps towards a better monitoring of dust burden at various spatiotemporal scales.

**Therefore, I think this analysis is not well conceived because the dataset in my opinion is not adequate for addressing the questions set out to answer in this study. My recommendation is to**

**reject the paper in this form. Below I suggest on one possible to use the same dataset and salvage this submission (so perhaps it can be labeled as "Major corrections and resubmit")**

**Suggestion for improvement of the manuscript**

**This dataset contains a valuable aggregation of satellite data and I think is still of value, particularly if merged with LIVAS. My advice to the author is to use this dataset to update some of the reference studies on dust activity using only satellite data. For example, the satellite data set can be used to update some of the Ginoux et al, (2012b) study regarding location of dust sources and activity. The database from this submission is more complete and is updated with respect what was used in the Ginoux paper. Such updated analysis will be very welcomed. For example, such analysis would include the observations of high latitude dust that are already available within the MODIS data base. They sources were not in the original Ginoux analysis nor in the Voss and Evan 2021 paper.**

We appreciate these interesting suggestions and nice ideas mentioned by the reviewer. We still believe that using a published and evaluated methodology with quantified uncertainties (Gkikas et al., 2021) to provide an update of the dust optical depth climatology globally with the given uncertainty and compare it with other approaches represents a relevant contribution to the literature.

Concerning the interesting ideas proposed by the reviewer here, we are indeed preparing a series of studies (some of them are ongoing and some of them are not yet at a mature level) related to model evaluation/constraint, impact on radiation fields and data assimilation. Among them, we are also working on merging the MIDAS and LIVAS dust optical depth datasets. Moreover, it is in our plans to utilize the MIDAS DOD product for identifying dust sources following Ginoux et al. (2012). Currently, we are testing the spatial distribution of FoOs by setting different DOD cutoff thresholds and then we will process ancillary data for the discrimination between natural and anthropogenic dust sources. In the figure below, it is illustrated the geographical distribution of FoOs (the percentage of DOD exceedances above 0.2), over North Africa and Middle East, based on our preliminary results, representative for the period 2003-2017.

**Percentage of days above the defined DOD threshold [ANNUAL]**

[Figure]

Frequency of DOD occurrence > 0.2

0.0   5.0   10.0   15.0   20.0   25.0   30.0   35.0   40.0   45.0   50.0   55.0   60.0   65.0   70.0

Data Min = 0.0, Max = 71.7

---

## Author Response (AR2)

Dear Editor,

I would like to thank you again for coordinating the review process of the acp-2021-572 manuscript.

The revised text has been modified according to your recommendations. More specifically, we have added the following paragraph in the conclusions section aiming to highlight all aspects regarding the uncertainty of the MIDAS DOD product.

*"Concerning the DOD uncertainties presented here, in the MIDAS dataset, MODIS AOD retrievals, obtained based on different assumptions in the respective algorithms, and MERRA-2 products are mixed. Therefore, the AOD and MDF errors, combined in the DOD uncertainty and carried through spatial and temporal averaging, are more likely heterogeneous and quite difficult to be quantified. Actually, the evaluation of spaceborne retrievals and numerical outputs can be much more complex and definitely further work is needed towards optimizing the confidence margins of total (speciated) optical depth levels. Quantifying accurately satellite based aerosol uncertainties is still an open issue and it is among our priorities to minimize the impacts of the aforementioned drawbacks and misrepresentations in the future versions of the MIDAS dataset."*

We hope that the revised manuscript addresses all the points raised by the two initial reviewers and it is suitable for publication in ACP in its current form. However, if you think that further clarifications or corrections are needed, then we can modify accordingly the text.

Finally, in the second round of comments, the Reviewer 3 mentioned that it is needed a much lengthier exchange regarding the calculation of the MIDAS DOD uncertainty. It would very useful if he/she would like to communicate with us in order to discuss further.

All the best,

Antonis Gkikas